# Embedding Trajectory for Out-of-Distribution Detection in Mathematical Reasoning

**Yiming Wang**[α] **Pei Zhang**[β,γ] **Baosong Yang**[β,✉] **Derek F. Wong**[γ]
**Zhuosheng Zhang**[α] **Rui Wang**[α,✉]

[α]Shanghai Jiao Tong University   [β]Tongyi Lab   [γ]NLP²CT Lab, University of Macau

✉: Corresponding Author
Email: [α]{yiming.wang, wangrui12}@sjtu.edu.cn
[β]yangbaosong.ybs@alibaba-inc.com

## Abstract

Real-world data deviating from the independent and identically distributed (*i.i.d.*) assumption of in-distribution training data poses security threats to deep networks, thus advancing out-of-distribution (OOD) detection algorithms. Detection methods in generative language models (GLMs) mainly focus on uncertainty estimation and embedding distance measurement, with the latter proven to be most effective in traditional linguistic tasks like summarization and translation. However, another complex generative scenario mathematical reasoning poses significant challenges to embedding-based methods due to its high-density feature of output spaces, but this feature causes larger discrepancies in the embedding shift trajectory between different samples in latent spaces. Hence, we propose a trajectory-based method TV Score, which uses trajectory volatility for OOD detection in mathematical reasoning. Experiments show that our method outperforms all traditional algorithms on GLMs under mathematical reasoning scenarios and can be extended to more applications with high-density features in output spaces, such as multiple-choice questions.

⭘ https://github.com/Alsace08/OOD-Math-Reasoning

## 1 Introduction

The rapid development of generative language models (GLMs) [40, 41, 5, 2, 50] has empowered them to fit diverse and challenging datasets, showing strong generalization over in-distribution (ID) test data satisfying the independent and identically distributed (*i.i.d.*) assumption. However, unconstrained inputs in real-world settings frequently trigger distributional drifts, called out-of-distribution (OOD) data. In such scenarios, model performance often deteriorates unexpectedly, yielding harmful outcomes. Thus, OOD detection [38, 4] is critical in safeguarding model security.

A highly scalable detection method must not rely on specific OOD data distributions, so simulating scenarios where OOD data is unavailable is more promising. Most existing research mainly focuses on vision and text classification tasks [12, 30, 55, 26, 48]. In contrast, studies addressing OOD detection on GLMs remain relatively niche, despite the more severe risks associated with OOD perils in GLMs due to the potential for error propagation in autoregressive generated sequences [43]. Existing methods on GLMs only focus on traditional text generation scenarios like summarization and translation, and these methods did not step outside the research framework of uncertainty estimation [33, 31] and embedding distance measure [43]. Among these, [43] has demonstrated that embedding-based methods are currently the only optimal solution for text generation scenarios, they determine whether a new sample is ID or OOD by calculating the *Mahalanobis Distance* [22] between the new sample embedding and ID embedding distribution in the static input or output space.

Recently, mathematical reasoning has emerged as a challenging generative task and a crucial benchmark for evaluating model abilities. However, it presents unique phenomena in both input and output spaces that render embedding-based methods inapplicable, as shown in Figure 1:

38th Conference on Neural Information Processing Systems (NeurIPS 2024).

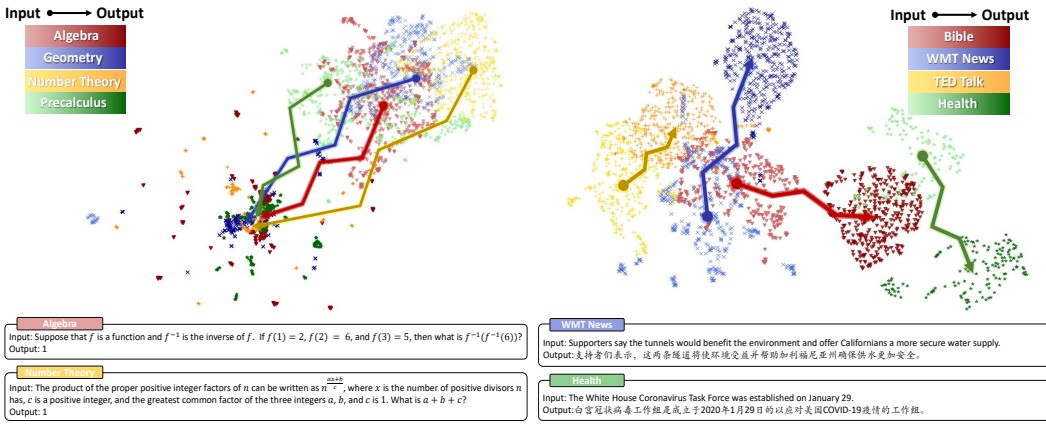

(a) **Mathematical Reasoning**  (b) **Text Generation (Translation)**

Figure 1: Embedding projection and cases of input and output spaces under mathematical reasoning and text generation scenarios. We select MATH [6] dataset for mathematical reasoning and OPUS [49] for text generation, each with four diverse domains. Different colors represent different domains, with lighter and darker shades indicating input and output. We use SimCSE [9] for sentence embeddings and UMAP [34] for dimensionality reduction. Appendix B shows detailed settings and examples.

- ***Input space*** *of mathematical reasoning exhibits vague clustering features across various domains, in contrast to the more defined clusters observed in text generation.* This indicates that embedding may struggle to capture the complexity of mathematical questions.
- ***Output space*** *of mathematical reasoning exhibits high-density characteristics with significant overlap between different domains.* we call this phenomenon **Pattern Collapse**. Since the output is mathematically symbolic [53, 37], it compresses the search space, increasing the likelihood of overlap between questions from disparate domains (As cases of Figure 1a). More importantly, the sequence tokenization used in GLMs allows for substantial token sharing among mathematically distinct expressions, as these mathematical tokens are primarily drawn from digits 0-9 and finite special symbols such as decimal points and square roots, rather than diverse linguistic elements. These scalar outputs lack distinctive features associated with specific domain distributions.

More discussion of the two phenomena, especially the "*pattern collapse*", will be presented in Section 6. Given the limitations of traditional methods, we aim to explore innovative OOD detection solutions in mathematical reasoning scenarios. We transform our focus from static embedding space to the dynamic embedding trajectory, this motivation stems from a theoretical insight, as shown in Figure 2: *"pattern collapse" causes the convergence of the trajectory endpoints of different samples, leading to significant trajectory differences across samples*. In Section 2.1, we model and prove this hypothesis to elucidate the intuition of using trajectories as a measure. Subsequently, in Section 2.2, we perform empirical experiments to investigate the underlying causes of trajectory differences between ID and OOD samples. Our findings reveal a phenomenon we term **Early Stabilization**, wherein *GLMs achieve primary reasoning in later stages for ID samples, a pattern not observed in OOD samples*. This observation provides direct evidence for the rationale behind using trajectory as a measure.

Based on these analyses, in Section 3, we propose the **T**rajectory **V**olatility detection algorithm (**TV Score**) for mathematical reasoning. Then in Section 4 and 5, we conduct extensive OOD detection experiments with diverse datasets and GLMs to validate the effectiveness of our method. We also tackle the challenging scenario of OOD quality estimation, which raises higher precision demands on the OOD scores. Results indicate that our method surpasses all traditional algorithms for GLMs under the mathematical reasoning scenario. Additionally, we demonstrate the extension of our method to a broader range of tasks with high-density features in output spaces, such as multiple-choice questions.

**Problem Statement: OOD Detection on GLMs.**    We start by formalizing GLMs. Let $\mathcal{X}$ and $\mathcal{Y}$ be the input and output spaces with $P_{\mathcal{X}}$ and $P_{\mathcal{Y}}$ be the marginal distributions for respective space, and $P_{\mathcal{X},\mathcal{Y}}$ is the joint data distribution defined over $\mathcal{X} \times \mathcal{Y}$. GLMs are trained given input sequence $\boldsymbol{x} = x_1x_2...x_t \sim P_{\mathcal{X}}$ of length $t$ to autoregressively generate the next token in the corresponding output sequence $\boldsymbol{y} = y_1y_2...y_n \sim P_{\mathcal{Y}}$ of length $n$ over the likelihood model $p_{\boldsymbol{\theta}}(\boldsymbol{y}|\boldsymbol{x}) = \prod_{i=1}^{n} p_{\boldsymbol{\theta}}(y_i|\boldsymbol{y}_{\prec i}, \boldsymbol{x})$, where each $x_i$ and $y_i$ are taken from vocabulary $\mathcal{V}$ and $\boldsymbol{\theta}$ is sampled from the parameter space $\Theta$.

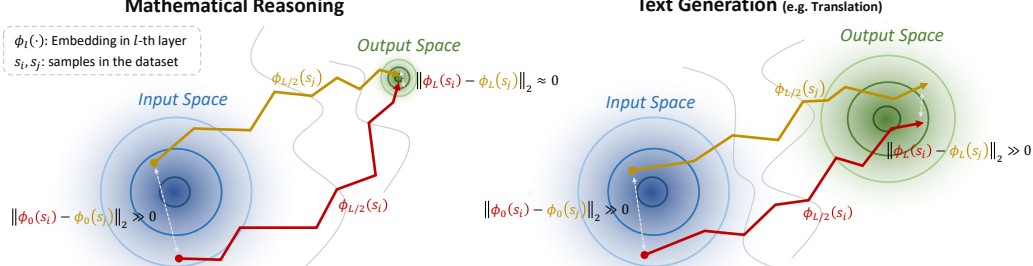

Figure 2: The "pattern collapse" phenomenon only exists in mathematical reasoning scenarios, where two samples initially distant in distance will converge approximately at the endpoint after undergoing embedding shifts, and does not occur in text generation scenarios. This produces a greater likelihood of trajectory variation under different samples in mathematical reasoning.

Assume that $\widetilde{P}_{\mathcal{X},\mathcal{Y}}$ denote a distribution sufficiently different from $P_{\mathcal{X},\mathcal{Y}}$, the goal of OOD detection in GLMs is to find a score function $f(\boldsymbol{x},\boldsymbol{y},\boldsymbol{\theta})$ for each sample and a threshold $\epsilon$, which may rely on the features of $\mathcal{X}$, $\mathcal{Y}$, and $\Theta$, to achieve a high discrimination accuracy goal:

$$\max_f \; \mathbb{P}_{(\boldsymbol{x},\boldsymbol{y})\sim P_{\mathcal{X},\mathcal{Y}}} \left[f(\boldsymbol{x},\boldsymbol{y},\boldsymbol{\theta}) < \epsilon\right] + \mathbb{P}_{(\widetilde{\boldsymbol{x}},\widetilde{\boldsymbol{y}})\sim\widetilde{P}_{\mathcal{X},\mathcal{Y}}} \left[f(\widetilde{\boldsymbol{x}},\widetilde{\boldsymbol{y}},\boldsymbol{\theta}) > \epsilon\right]. \tag{1}$$

## 2 Dynamic Embedding Trajectory Distinguishes ID and OOD Samples in Mathematical Reasoning

**We begin by defining the embedding trajectory and its volatility features.** Given a sample $s$, we put its input to the language model $p_\theta$ with $L$ layers, and the output sequence consists of $T$ tokens. For the $t$-th token, we denote its output embedding at layer $l$ as $\boldsymbol{h}_l^t$, with each embedding a $d$-dimensional vector. Following the definitions in [43, 51], we define the average embedding $\boldsymbol{y}_l = \frac{1}{T}\sum_{t=1}^{T}\boldsymbol{h}_l^t$ as the sentence embedding at layer $l$. Then, the embedding trajectory is formed as a progressive chain of these embeddings: $\boldsymbol{y}_0 \to \boldsymbol{y}_1 \to \cdots \to \boldsymbol{y}_l \to \cdots \to \boldsymbol{y}_{L-1} \to \boldsymbol{y}_L$. To measure the change magnitudes of the embedding trajectory in the latent space, we define two types of trajectory volatilities:

- *Dimension-independent volatility* $\boldsymbol{V}_I(s) \in \mathbb{R}^d$: For sample $s$, we first obtain the embedding difference vector $|\boldsymbol{y}_l - \boldsymbol{y}_{l-1}|$ between each adjacent-layer pair $l-1$ and $l$. $\boldsymbol{V}_I$ is the average of all differences across layers, it captures the local trajectory changes across individual dimensions:

$$\boldsymbol{V}_I(s) = \frac{1}{L}\cdot\sum_{l=1}^{L}|\boldsymbol{y}_l - \boldsymbol{y}_{l-1}| = \frac{1}{L}\cdot\sum_{l=1}^{L}\left(\left|y_l^1 - y_{l-1}^1\right|, \cdots, \left|y_l^d - y_{l-1}^d\right|\right)^\top. \tag{2}$$

- *Dimension-joint volatility* $V_J(s) \in \mathbb{R}$: For sample $s$, we first obtain the L2-norm of the embedding difference $||\boldsymbol{y}_l - \boldsymbol{y}_{l-1}||_2$ between each adjacent-layer pair $l-1$ and $l$. $V_J$ is the average of all such differences across layers, it captures the global changes in the trajectory:

$$V_J(s) = \frac{1}{L}\cdot\sum_{l=1}^{L}||\boldsymbol{y}_l - \boldsymbol{y}_{l-1}||_2 = \frac{1}{L}\cdot\sum_{l=1}^{L}\sqrt{\frac{1}{d}\cdot\sum_{i=1}^{d}\left(y_l^i - y_{l-1}^i\right)^2}. \tag{3}$$

In this section, we will clarify our motivation: *Why Dynamic Embedding Trajectory As The Measure?* This stems from the phenomenon of "*pattern collapse*" in the output space, where the endpoints of different sample trajectories converge to a high-density region. This constraint potentially increases the likelihood of trajectory differences across samples. We will model and prove this theoretical intuition in Section 2.1 through the *Dimension-independent volatility*. After gaining insight into trajectory differences, we specifically explore the differences between ID and OOD sample trajectories. We will empirically investigate this in Section 2.2 through the *Dimension-joint volatility*.

### 2.1 Theoretical Intuition: Trajectory Differences with Higher Likelihood

Figure 1 has illustrated the "*pattern collapse*" phenomenon in the output space in mathematical reasoning scenarios. We abstract this phenomenon in Figure 2, which compares the trajectory

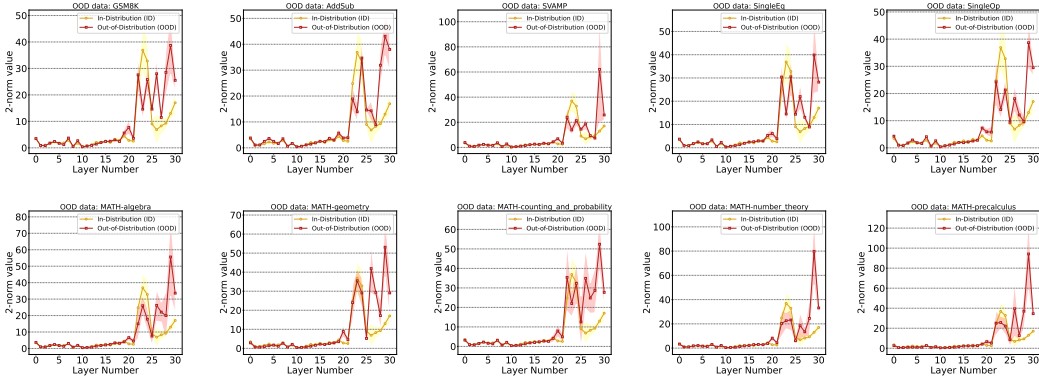

Figure 3: Trajectory volatility curve comparisons between one ID data and ten OOD data from diverse mathematical domains. Each trajectory represents the average of all samples from the corresponding datasets, with color shading being the sample standard deviation. Llama2-7B is used for the backbone.

trend between different samples in the mathematical reasoning and text generation scenarios: In mathematical reasoning, when initial points of two trajectories are separated by any distance in the input space, they typically converge to a significantly closer distance in the output space after undergoing an embedding shift. However, in text generation, outputs from different samples may not exhibit this same convergence. This finding inspires the following theoretical intuition: **Hard constraints on trajectory endpoints in mathematical reasoning allow for a higher probability of trajectory differences under different samples**, as expressed by the key hypothesis:

**Hypothesis 1** *In scenarios characterized by pattern collapse in the output space, the probability of trajectory volatility differences across samples increases.*

We now formalize this hypothesis. Assume that the output space is a Gaussian distribution $\mathcal{N}(c, \Sigma^2)$, so the output embedding $y_L \sim \mathcal{N}(c, \Sigma^2)$. For two different samples $s_i$ and $s_j$, their dimension-independent volatilities are $V_I(s_i)$ and $V_I(s_j)$, respectively. We want to show that when pattern collapse exists, the probability of $V_I(s_i) - V_I(s_j) \neq 0$ will increase. According to the pattern collapse features under different scenarios, we can constraint the $y_L$ as follows:

- For mathematical reasoning with pattern collapse, $\Sigma \to O$, so we approximate that $y_L \equiv c$;
- For text generation without pattern collapse, $\Sigma = \mathrm{diag}(\delta_1, \delta_2, \cdots, \delta_D) \neq O$, so $y_L \not\equiv c$.

With such formalized constraints, we model our Hypothesis 1 as a main theorem:

**Theorem 2.1 (Main Theorem)** *We assume that $\{y_l\}_{l=1}^L$ are all independent variables sampling from vector space $\mathbb{R}^d$. For different samples $s_i$ and $s_j$, their embedding sets are $\{[y_i]_l\}_{l=1}^L$ and $\{[y_j]_l\}_{l=1}^L$, respectively. The likelihood of trajectory volatility differences between $s_i$ and $s_j$ under mathematical reasoning scenarios is higher than that under text generation scenarios, which means:*

$$\mathbb{E}_{\{[y_i]_l\}_{l=1}^L, \{[y_j]_l\}_{l=1}^L \sim \mathcal{U}(\mathbb{R}^d)} \left\{ V_I(s_i) - V_I(s_j) \neq 0 | [y_i]_L, [y_j]_L \equiv c \right\}$$
$$> \mathbb{E}_{\{[y_i]_l\}_{l=1}^L, \{[y_j]_l\}_{l=1}^L \sim \mathcal{U}(\mathbb{R}^d)} \left\{ V_I(s_i) - V_I(s_j) \neq 0 | [y_i]_L, [y_j]_L \sim \mathcal{N}(c, \Sigma^2) \right\}$$

Due to space limits, we present the **Complete Proof** in Appendix C. This demonstrates that when pattern collapse occurs, the probability of trajectory volatility differences across samples increases.

## 2.2 Empirical Analysis: Early Stabilization of ID samples

The theoretical analysis in Section 2.1 provides an intuition for using embedding trajectory as the measure for distinguishing between different samples. However, the specific trajectory differences between ID and OOD samples remain unclear. Thus, we investigate this empirically in this section.

We select eleven mathematical reasoning datasets: one for ID data and ten for OOD data. The MultiArith dataset serves as the ID data, while the OOD datasets include GSM8K, SVAMP, AddSub, SingleEq, SingleOp, and MATH. The latter encompasses five tasks across various mathematical domains: Algebra, Geometry, Counting and Probability, Number Theory, and Precalculus. This selection includes varying task types and levels of difficulty. We first train a base Llama2-7B (32 layers) using the ID training set, then inference on the ID test set and all OOD test sets. Details regarding the datasets, model, and implementation can be found in Section 4.1.

We measure the dimension-joint volatility for all samples from each ID and OOD test set. Specifically, we compute each adjacent-layer change magnitude $||\boldsymbol{y}_l - \boldsymbol{y}_{l-1}||_2$, and connect all values as a change curve, with higher value means the higher volatility. Figure 3 shows ten curve comparisons, with each sub-figure including the ID data and one OOD data. We find that the change magnitude slightly until the 20th layer. After 20 layers, for ID data, the change magnitude is again suppressed after a few layers of inference, while for OOD data, the magnitude is maintained at a relatively high level.

We term this phenomenon "**Early Stabilization**": For ID data, GLMs largely complete their reasoning in the mid-to-late stages, and simple adjustments are sufficient after that. However, for OOD data, GLMs can still not complete accurate reasoning at a later stage. They thus can only randomly switch to a specific output pattern, *i.e.*, the scalar mathematical expression pattern. This provides strong evidence that using embedding trajectory for OOD detection may be effective.

## 3 TV Score: Trajectory Volatility Score for OOD Detection

In Section 2.2, we have identified a significant difference in embedding trajectory volatilities between ID and OOD samples in mathematical reasoning. We now aim to leverage this phenomenon to develop a lightweight OOD detection solution tailored for mathematical reasoning scenarios.

Inspired by static embedding methods that use the ID sample embedding cluster as the reference to measure the *Mahalanobis Distance* (MaDis) [22], we similarly aim to use the ID sample trajectory cluster as the reference to measure the difference between them and a new sample trajectory. While measuring the difference between an embedding and an embedding cluster only requires a simple MaDis calculation, quantifying the difference between a trajectory and a trajectory cluster is less intuitive. Thus, we seek a transition idea starting from the static space Gaussian assumption.

First, we obey the assumption under static embeddings [3, 43] to fit all ID embeddings at each layer $l$ to a Gaussian distribution $\mathcal{G}_l = \mathcal{N}(\boldsymbol{\mu}_l, \boldsymbol{\Sigma}_l)$. Next, for a new sample with $\boldsymbol{y}_l$ be its embedding at layer $l$, we map it to its MaDis $f(\boldsymbol{y}_l) : \mathbb{R}^d \to \mathbb{R}$ with $\mathcal{G}_l$ as follows:

$$f(\boldsymbol{y}_l) = (\boldsymbol{y}_l - \boldsymbol{\mu}_l)^\top (\Sigma_l)^{-1} (\boldsymbol{y}_l - \boldsymbol{\mu}_l) \quad (0 \le l \le L). \tag{4}$$

Finally, we treat the MaDis differences $|f(\boldsymbol{y}_l) - f(\boldsymbol{y}_{l-1})|$ between adjacent layer-pairs $l-1$ and $l$ as the adjacent-layer volatility of the new sample, and average all adjacent-layer volatilities as the final trajectory volatility score (TV Score $S$) of this sample:

$$S = \frac{1}{L} \cdot \sum_{l=1}^{L} |f(\boldsymbol{y}_l) - f(\boldsymbol{y}_{l-1})| . \qquad \textbf{(TV Score)}$$

The anticipated trend is that when the two adjacent embeddings after MaDis mapping exhibit a greater difference, the embedding change between the two layers is more volatile compared to ID data. This enables us to identify new samples with trajectory volatility features that significantly deviate from those of the ID samples, thus increasing the likelihood that they are OOD samples.

Furthermore, as a trajectory, outliers in the trajectory may significantly impact feature extraction [21, 28]. To mitigate this, we explore higher-order differential smoothing techniques to enhance trajectory smoothness. We first define the $k$-order embedding $\nabla^k \boldsymbol{y}_l$ and Gaussian distribution $\nabla^k \mathcal{G}_l = \nabla^k \mathcal{N}(\boldsymbol{\mu}_l, \boldsymbol{\Sigma}_l)$ for all $l \le L - k$ based on the backward difference:

$$\nabla^k \boldsymbol{y}_l = \sum_{i=0}^{k} (-1)^{k+i} \mathrm{C}_k^i \boldsymbol{y}_{l+k}, \quad \nabla^k \mathcal{G}_l = \mathcal{N} \left( \sum_{i=0}^{k} (-1)^{k+i} \mathrm{C}_k^i \boldsymbol{\mu}_{l+k}, \sum_{i=0}^{k} \mathrm{C}_k^i \boldsymbol{\Sigma}_{l+k} \right), \tag{5}$$

where $\mathrm{C}_k^i = \frac{k!}{i!(k-i)!}$. Similarly, we mapping $\nabla^k \boldsymbol{y}_l$ to its MaDis $\nabla^k f(\boldsymbol{y}_l) : \mathbb{R}^d \to \mathbb{R}$ with $\nabla^k \mathcal{G}_l$:

$$\nabla^k f(\boldsymbol{y}_l) = \left( \nabla^k \boldsymbol{y}_l - \sum_{i=0}^{k} (-1)^{k+i} \mathrm{C}_k^i \boldsymbol{\mu}_{l+k} \right)^\top \left( \sum_{i=0}^{k} \mathrm{C}_k^i \boldsymbol{\Sigma}_{l+k} \right)^{-1} \left( \nabla^k \boldsymbol{y}_l - \sum_{i=0}^{k} (-1)^{k+i} \mathrm{C}_k^i \boldsymbol{\mu}_{l+k} \right). \tag{6}$$

Following the definition of TV Score, we define the trajectory volatility score after differential smoothing (TV Score w/ DiSmo $\nabla^k S$) as follows:

$$\nabla^k S = \frac{1}{L} \cdot \sum_{l=1}^{L} \left| \nabla^k f(\boldsymbol{y}_l) - \nabla^k f(\boldsymbol{y}_{l-1}) \right| . \qquad \textbf{(TV Score w/ Dismo)}$$

The **Algorithmic Process** and **Computational Complexity** are detailed in Appendix D.

# 4 Experiments

## 4.1 Setup

**Dataset Selection.** For the ID dataset, we use the MultiArith [44], which consists of Math Word Problems on arithmetic reasoning. For the OOD datasets, we intuitively introduce two types of detection scenarios following [43]: (i) *Far-shift OOD* scenario, we select the MATH [11] dataset with five domains of algebra, geometry, counting and probability, number theory, and precalculus; (ii) *Near-shift OOD* scenario, we select five independent datasets: GSM8K [6], SVAMP [39], AddSub [13], SingleEq [18], and SingleOp [18]. We consider the ID data negative(-) and the OOD data positive(+). Refer to Appendix E.1 for basic information and OOD features of these datasets.

**Data Split and Sampling.** Given the limited data size of MultiArith, totaling only 600 samples and lacking a standard division, we allocate 360 samples for training and 240 for testing. However, with such a small test set, randomness in evaluation becomes a concern. To mitigate this, we conduct test sampling and set the sampling size as 1000. Specifically, we denote ID dataset as $\mathcal{D}_{\text{in}}$ and OOD dataset as $\mathcal{D}_{\text{out}}$. For each sampling, the collection is $\{\mathcal{D}_{\text{in}}, \widetilde{\mathcal{D}}_{\text{out}}\}$ where $\widetilde{\mathcal{D}}_{\text{out}} \subset \mathcal{D}_{\text{out}}$ and $|\mathcal{D}_{\text{in}}| = |\widetilde{\mathcal{D}}_{\text{out}}|$, this guarantees positive and negative sample balance. We report both the mean and standard variance of the results to enhance the reliability of evaluations. Refer to Appendix E.2 for the ID dataset split.

**Implementation.** *To measure the application value of our method used in cutting-edge GLMs*, we use Llama2-7B [50] and GPT2-XL (1.5B) [5] as our backbones for ID dataset training. Refer to Appendix E.3 for training details. However, there exists uncertainty about the data used in the pre-training phase, especially for Llama2 because its data is closed-source. Some research [57, 59] have confirmed the absence of data leakage in Llama2 for the MATH and GSM8K datasets, we still conduct pre-experiments to examine the rationality of the OOD data selection rigorously. Our criterion is the claim that a dataset can be categorized as OOD if it exceeds the capabilities of the base model, as proposed in prior studies [47, 27]. Results of the pre-experiments are shown in Appendix E.4, and they can confirm that these datasets can be considered as OOD data for the two GLMs.

**Baseline.** We compare our method with five training-free baselines where OOD training data are unavailable. We refer to the latest survey [20] to select them for the scarcity of OOD detection methods on GLMs: (1) Maximum Softmax Probability (Prob.) [12]; (2) Monte-Carlo Dropout [8]; (3) Sequence Perplexity [3]; (4) Input Embedding [43]; (5) Output Embedding [43]. Refer to Appendix E.5 for details. Additionally, we set the smoothing order $k$ ranges from 1 to 5 for TV Score w/ DiSmo and report the highest among them when with smoothing.

**Evaluation.** We divide the OOD detection evaluation into two scenarios: (1) **Offline Detection**, which classifies whether samples from a given list belong to OOD. For each collection $\{\mathcal{D}_{\text{in}}, \widetilde{\mathcal{D}}_{\text{out}}\}$, we report the AUROC [35] and FPR95 metrics. The former represents the area under the ROC curve and the latter represents the value of FPR at 95% TPR. (2) **Online Detection**, which utilizes the offline detection results to calculate an optimal classification threshold for a direct determination of whether new samples belong to OOD. We introduce the metrics in the corresponding result sub-sections.

## 4.2 Main Results

**Offline Detection.** Table 1 presents the results of the offline detection scenarios.

- *Performance Analysis*: In the far-shift OOD setting, **our average performance surpasses 98 (Llama2-7B) and 96 (GPT2-XL) under the AUROC metric**, surpassing the optimal baseline by 10+ points. Moreover, our performance stands at **an impressive 5.21 (Llama2-7B) and 9.89 (GPT2-XL) under the FPR95 metric, representing a remarkable 80%+ reduction** compared to the optimal baseline, far surpassing all baseline methods. In the near-shift OOD setting, the robustness of our method is even more impressive. All of the baseline methods show significant performance degradation, especially in Llama2-7B, with the AUROC metric decreasing below 60 and the FPR95 elevating above 80. However, our method maintains excellent performances, with AUROC scores surpassing 90 and FPR95 below 30. This indicates that for more fine-grained OOD detection scenarios, our method demonstrates greater adaptability.

- *Model Analysis*: Comparing performances of Llama2-7B and GPT2-XL, we find two phenomena: (i) Results on GPT2-XL are more stable, performance differences between GPT2-XL on far- and near-drift settings are not significant, while Llama2-7B shows a significant performance

Table 1: AUROC and FPR95 results of the **Offline Detection** scenario. Underline and **bold** denote SOTA among all baselines and all methods, respectively. We report the *average* results under each setting in the main text, results of each dataset are shown in Table 11 and 12 (Appendix F).

| Model | **Llama2-7B** [50] | | | | **GPT2-XL** [5] | | | |
|---|---|---|---|---|---|---|---|---|
| *Metric* | Far-shift OOD | | Near-shift OOD | | Far-shift OOD | | Near-shift OOD | |
| *Method* | AUROC ↑ | FPR95 ↓ | AUROC ↑ | FPR95 ↓ | AUROC ↑ | FPR95 ↓ | AUROC ↑ | FPR95 ↓ |
| Max Softmax Prob. [12] | $78.66_{\pm1.38}$ | $81.44_{\pm3.56}$ | $60.14_{\pm1.54}$ | $88.91_{\pm2.41}$ | $70.54_{\pm1.42}$ | $78.29_{\pm2.02}$ | $67.12_{\pm1.20}$ | $76.27_{\pm2.66}$ |
| Monte-Carlo Dropout [8] | $68.63_{\pm2.21}$ | $87.04_{\pm4.88}$ | $52.33_{\pm2.21}$ | $91.92_{\pm1.89}$ | $66.18_{\pm1.87}$ | $84.69_{\pm1.65}$ | $63.54_{\pm1.72}$ | $78.08_{\pm2.50}$ |
| Perplexity [3] | $\underline{85.64}_{\pm1.46}$ | $\underline{53.06}_{\pm4.36}$ | $59.35_{\pm1.89}$ | $86.09_{\pm1.89}$ | $80.82_{\pm1.04}$ | $64.53_{\pm2.10}$ | $73.74_{\pm1.12}$ | $72.39_{\pm1.27}$ |
| Input Embedding [43] | $75.89_{\pm1.03}$ | $67.87_{\pm3.69}$ | $\underline{60.33}_{\pm1.37}$ | $\underline{84.65}_{\pm2.53}$ | $\underline{86.26}_{\pm0.84}$ | $\underline{49.33}_{\pm2.10}$ | $\underline{83.22}_{\pm0.88}$ | $\underline{52.90}_{\pm3.16}$ |
| Output Embedding [43] | $74.86_{\pm1.39}$ | $75.21_{\pm2.16}$ | $44.50_{\pm1.06}$ | $86.46_{\pm1.59}$ | $77.95_{\pm1.16}$ | $65.64_{\pm3.42}$ | $79.28_{\pm1.24}$ | $64.70_{\pm2.72}$ |
| TV Score (Ours) | $\mathbf{98.76}_{\pm0.11}$ | $\mathbf{5.21}_{\pm0.98}$ | $\mathbf{92.64}_{\pm0.39}$ | $\mathbf{28.39}_{\pm1.38}$ | $93.47_{\pm0.08}$ | $24.10_{\pm0.95}$ | $\mathbf{94.86}_{\pm0.23}$ | $13.82_{\pm0.36}$ |
| w/ DiSmo (Ours) | $93.25_{\pm0.76}$ | $41.82_{\pm4.69}$ | $56.99_{\pm1.41}$ | $88.01_{\pm1.71}$ | $\mathbf{96.54}_{\pm0.11}$ | $\mathbf{9.89}_{\pm0.61}$ | $94.19_{\pm0.25}$ | $\mathbf{13.66}_{\pm0.69}$ |
| Δ (**bold** - underline) | +13.12 | -47.85 | +32.31 | -56.26 | +10.28 | -39.44 | +11.64 | -39.24 |

Table 2: Accuracy and Robustness results of the **Online Detection** scenario. We mainly compare our method with embedding-based methods, and **bold** denotes the best among these methods.

| | Far-shift OOD Setting | | | Near-shift OOD Setting | | |
|---|---|---|---|---|---|---|
| *Dataset* | **Accuracy↑** | **Robustness↓** | *Dataset* | **Accuracy↑** | **Robustness↓** | |
| | I-Emb. / O-Emb. / TV (ours) | | | I-Emb. / O-Emb. / TV (ours) | | |
| Algebra | 76.43 / 45.42 / **93.88** | 5.27 / 6.94 / **0.97** | GSM8K | 81.49 / 75.32 / **93.39** | 10.08 / 3.36 / **2.05** | |
| Geometry | 74.32 / 54.79 / **94.47** | 2.44 / 2.43 / **1.65** | SVAMP | 68.66 / 63.33 / **94.88** | 5.26 / 3.54 / **2.13** | |
| Cnt.&Prob | 50.31 / 27.55 / **93.74** | 9.99 / **2.34** / 2.36 | AddSub | **79.16** / 78.09 / 74.11 | 3.21 / 6.98 / **2.77** | |
| Num.Theory | 85.80 / 54.38 / **92.08** | 3.31 / 11.45 / **2.34** | SingleEq | 59.83 / 72.56 / **93.15** | 11.57 / **3.14** / 3.17 | |
| Precalculus | 80.33 / 88.50 / **99.28** | 6.13 / 1.38 / **0.67** | SingleOp | 69.38 / 62.20 / **95.75** | 4.00 / **2.37** / 2.45 | |
| *Average* | 73.44 / 54.13 / **94.69** | 5.43 / 4.91 / **1.60** | *Average* | 71.70 / 70.30 / **90.26** | 6.82 / 3.88 / **2.51** | |

degradation (mainly for baselines) on near-shift setting; (ii) the DiSmo technique is more effective on GPT2-XL, which suggests that there are more anomalous learning tendencies in latent spaces of small models, and the smoothing helps to minimize these anomalies.

In addition, we conduct significant tests (Details are shown in Table 11 - 14). We find that our methods almost pass all significance tests, while the embedding-based methods have the lowest pass rate among baselines, suggesting that their results are more susceptible to sampling error. We also find that the performance of differential smoothing fluctuates greatly in different settings. Therefore, we conduct the ablation of the smoothing order $k$. Results and analyses are shown in Appendix F.1.

**Online Detection.** In this part, we utilize the TV score for online OOD discrimination. For each collection $\{\mathcal{D}_{\text{in}}, \widetilde{\mathcal{D}}_{\text{out}}\}$, we obtain a detector and computer the optimal cut-off $\tau_i$ of Youden Index, which is at the point in the AUROC curve where $\text{TPR} - \text{FPR}$ is maximum. Then for all OOD samples $s \in \mathcal{D}_{\text{out}} - \widetilde{\mathcal{D}}_{\text{out}}$, we donate $t$ as the sampling size and computer the discrimination accuracy:

$$\text{Accuracy} = \frac{1}{t}\sum_{i=1}^{t} \frac{\sum_{s \in \mathcal{D}_{\text{out}} - \widetilde{\mathcal{D}}_{\text{out}}} \mathbb{I}\left[\text{TV-Score}(s) \geq \tau_i\right]}{\left|\mathcal{D}_{\text{out}} - \widetilde{\mathcal{D}}_{\text{out}}\right|}. \tag{7}$$

In addition, the discrimination accuracy should vary less under different data collections, reflecting the discriminator's robustness. Therefore, we denote the Robustness metric as sampling variance.

Table 2 presents the results in Llama2-7B. Compared to the embedding-based methods, **our TV score obtains about an average of 20-point accuracy improvement in both far-shift OOD and near-shift OOD settings**, and on some datasets, such as Cnt.&Prob, our TV score achieves more than 40 points of improvement. These all imply that TV Score can perform online discrimination of OOD samples more accurately. In addition, our TV score also possesses stronger robustness, which means that in real scenarios, we can find the optimal threshold more consistently in the face of different accessible ID and OOD data, reducing the potential riskiness due to uncontrollable data acquisition.

# 5 Generalizability Exploration

## 5.1 Beyond Detection: OOD Quality Estimation

In this part, we utilize the TV score for generative quality estimation (QE). For text generation, the QE performance is usually measured by calculating the correlation coefficient between automatic scores and human ratings. However, QE in mathematical reasoning scenarios is not a well-defined problem. For one mathematical question, its answer is either right or wrong, the intermediate state does not exist. For example, when the correct answer is 12.5, it is difficult to judge which is better between generated answers of 1.25 and 13.6. The human approach may be to judge by comparing the difference value or similarity like Rouge [24] and BertScore [60] between the generated answer and the correct answer, which is unfair to the machine because there is a lot of randomness in the intermediary process of computation, and the solution pattern of machines is case-based [14], so it is not suitable to judge the machine-generated results with customized mathematical rules.

Therefore, we use the binary direct matching[1] to compare the model-generated answers with the correct answers. Considering the open-ended output of the GLMs, we give a loose matching condition, *i.e.*, as long as the correct answer is included in the generated answer by the model, the generated answer is recognized as correct and the matching score is 1, otherwise the matching score is 0. We compute the Kendall rank correlation coefficient $\tau$ [45] and Spearman rank correlation coefficient [36] between each OOD score and the matching score.

Table 3: **OOD Quality Estimation**: Kendall's $\tau$ and Spearman correlation between various OOD scores and benchmark quality metric binary matching. Each column shows the correlation when ID and OOD samples are merged. Underline denotes the SOTA among all baselines, and **bold** denotes the SOTA among our methods. We report the *average* results under each setting in the main text, results of each dataset are shown in Table 13 and 14 (Appendix F).

| Model | Llama2-7B [50] | | | | GPT2-XL [5] | | | |
|---|---|---|---|---|---|---|---|---|
| Metric | Far-shift OOD | | Near-shift OOD | | Far-shift OOD | | Near-shift OOD | |
| Method | Kendall ↑ | Spearman ↑ | Kendall ↑ | Spearman ↑ | Kendall ↑ | Spearman ↑ | Kendall ↑ | Spearman ↑ |
| Max Softmax Prob. | $0.024_{\pm 0.020}$ | $0.038_{\pm 0.020}$ | $0.038_{\pm 0.018}$ | $0.026_{\pm 0.018}$ | $\underline{0.066}_{\pm 0.015}$ | $0.044_{\pm 0.016}$ | $\underline{0.057}_{\pm 0.018}$ | $0.057_{\pm 0.022}$ |
| Perplexity | $0.050_{\pm 0.015}$ | $0.045_{\pm 0.016}$ | $\underline{0.074}_{\pm 0.017}$ | $0.050_{\pm 0.018}$ | $0.036_{\pm 0.014}$ | $0.038_{\pm 0.017}$ | $0.035_{\pm 0.018}$ | $0.058_{\pm 0.019}$ |
| Input Embedding | $\underline{0.078}_{\pm 0.016}$ | $\underline{0.102}_{\pm 0.017}$ | $0.036_{\pm 0.018}$ | $\underline{0.115}_{\pm 0.017}$ | $0.059_{\pm 0.012}$ | $\underline{0.098}_{\pm 0.016}$ | $0.012_{\pm 0.018}$ | $\underline{0.068}_{\pm 0.016}$ |
| Output Embedding | $0.058_{\pm 0.018}$ | $0.025_{\pm 0.017}$ | $0.038_{\pm 0.015}$ | $0.012_{\pm 0.017}$ | $0.050_{\pm 0.012}$ | $0.016_{\pm 0.017}$ | $0.036_{\pm 0.017}$ | $0.029_{\pm 0.021}$ |
| TV Score (Ours) | $\mathbf{0.161}_{\pm \mathbf{0.012}}$ | $0.147_{\pm 0.015}$ | $\mathbf{0.159}_{\pm \mathbf{0.017}}$ | $\mathbf{0.158}_{\pm \mathbf{0.017}}$ | $0.138_{\pm 0.010}$ | $0.123_{\pm 0.013}$ | $\mathbf{0.131}_{\pm \mathbf{0.015}}$ | $0.146_{\pm 0.015}$ |
| w/ DiSmo (Ours) | $0.111_{\pm 0.016}$ | $\mathbf{0.152}_{\pm \mathbf{0.015}}$ | $0.113_{\pm 0.018}$ | $0.134_{\pm 0.017}$ | $\mathbf{0.139}_{\pm \mathbf{0.009}}$ | $\mathbf{0.141}_{\pm \mathbf{0.014}}$ | $0.123_{\pm 0.014}$ | $\mathbf{0.154}_{\pm \mathbf{0.016}}$ |
| $\Delta$ (**bold** - underline) | +0.083 | +0.050 | +0.085 | +0.043 | +0.073 | +0.043 | +0.074 | +0.086 |

Table 3 presents the results. For Llama2-7B, when compared with Kendall correlation, the correlation improvement of TV scores **over SOTA baselines reaches up to 100% under both far-shift and near-shift OOD settings.** Compared with Spearman correlation, TV scores demonstrate a correlation enhancement **over SOTA baselines by up to 100% under far-shift OOD setting and 30% under near-shift OOD setting**. GPT2-XL also demonstrates excellent performance. These findings indicate that our TV scores not only facilitate the binary discrimination of ID and OOD samples but also substantially reflect the quality and precision of generated mathematical reasoning.

## 5.2 Beyond Mathematical Reasoning

Apart from mathematical reasoning, our method also has a wider range of potential applications that **can be extended to any task where the output space exhibits the pattern collapse property**. An example would be multiple-choice questions, which is a popular evaluation tool in the era of large language models and also display the pattern collapse property due to the limited output space being confined to the "ABCD" four options. To verify the generalizability of our method, we conduct experiments using the multiple-choice dataset MMLU [10], and our method also outperforms all traditional algorithms in this setting. Results and analyses are shown in Appendix F.4.

---

[1]Although direct matching is the most accurate solution, it suffers from two issues: (i) Generated answers may include much noisy content, increasing the matching difficulty; (ii) Performances on GLMs of mathematical reasoning is poor, which unbalances the positive and negative samples and increases the randomness.

# 6 Rethink Inapplicability of Static Embedding Methods

In Figure 1, we find that the static embedding space for both input and output fails to capture distinct features across different domains. This renders traditional embedding-based methods unsuitable for mathematical reasoning tasks. Experimental results in Section 4 also verify this disadvantage. In this section, we conduct a detailed analysis of these phenomena, and reveal the root cause of the poor performance of static embedding methods in mathematical reasoning scenarios.

## 6.1 Input Space: Representation Dilemma on Mathematical Expressions

The input space in mathematical reasoning exhibits vague clustering features compared to text generation scenarios as shown in Figure 1. We speculate that as semantic representations, embeddings cannot accurately measure mathematical expressions in the mathematical sense.

We begin by considering a counter-intuitive toy example. We construct a benchmark expression "A: 2+3+4=?" and three diverse expressions: "X: 8-0+9=?", "Y: x^2*y^3*x^4=?", and "Z: 234*2+345*4+456*4-243*3=?". From the mathematical sense, X is ID for A, whereas Y and Z are OOD for A since the difficulty and knowledge used to solve Y and Z are largely different from A compared to X. Table 4 presents the cosine similarity between each of them and A. Notably, X exhibits the lowest similarity to A, primarily due to having fewer token sharing with A. This is inconsistent with human perception of mathematics.

Table 4: A toy example about cosine similarities between different mathematical expressions.

| Mathematical Expression | Cosine with Benchmark |
|---|---|
| Benchmark: 2+3+4=? | |
| 8-0+9=? | 0.65 |
| x^2*y^3*x^4=? | 0.83 |
| 234*2+345*4+456*4-243*3=? | 0.78 |

Table 5: AUROC score matrix produced after alternating the MATH dataset's five domains as ID and OOD data measured by **(a) Input Embedding Mahalanobis Distance** and **(b) Output Embedding (w/ CoT) Mahalanobis Distance**. Darker colors represent better performances.

| | | OOD Domain | | | | | | | | | |
|---|---|---|---|---|---|---|---|---|---|---|---|
| | | **(a) Input Embedding MaDis** | | | | | **(b) Output Embedding (w/ CoT) MaDis** | | | | |
| | | algebra | geometry | cnt.&prob | num.theory | precalculus | algebra | geometry | cnt.&prob | num.theory | precalculus |
| ID Domain | algebra | - | 63.77 | 80.01 | 58.79 | 50.80 | - | 71.37 | 84.24 | 65.06 | 56.71 |
| | geometry | 85.68 | - | 88.14 | 86.55 | 69.03 | 72.12 | - | 92.42 | 86.08 | 70.89 |
| | cnt.&prob | 49.95 | 44.94 | - | 38.02 | 51.76 | 46.19 | 53.61 | - | 40.15 | 51.63 |
| | num.theory | 66.35 | 78.87 | 50.00 | - | 71.93 | 63.02 | 78.76 | 69.85 | - | 67.96 |
| | precalculus | 85.14 | 79.31 | 86.18 | 89.34 | - | 79.84 | 85.65 | 86.77 | 90.35 | - |

To demonstrate this phenomenon, we use one of five domains from the MATH dataset as the ID and the remaining four domains as OOD. We rotate this setting across five iterations. Under each ID-OOD pair, we allocate half of the ID samples to compute the ID embedding distribution, while the other half and all OOD samples constitute the test samples. The Mahalanobis distance is then calculated between embeddings and the ID distributions for each sample input, yielding the OOD score. We employ SimCSE [9] to generate sentence embeddings. Table 5(a) presents the AUROC score matrix, in nearly half of the settings, the AUROC value hovers around 50, *i.e.*, completely random. This suggests that embeddings can not distinguish between domains in a mathematical sense.

## 6.2 Output Space: High-density "*Pattern Collapse*"

The output space in mathematical reasoning exhibits a unique high-density characteristic, which we call "*pattern collapse*". As shown in Figure 1, the distribution of output embeddings across various domains in the mathematical reasoning scenarios is significantly concentrated and indistinguishable.

Two reasons cause this phenomenon: **(1) Expression level**: Output space of mathematical reasoning is scalar [37], resulting in a compressed search space with a higher probability of overlap in two widely divergent questions. For example, "$1 + 3 =$" and "$\int_1^3 x \mathrm{d}x =$" are both "4"; **(2) Token level**: We usually categorize different mathematical reasoning domains from a mathematical sense. However, *GLMs model real numbers or mathematical expressions not in a mathematical sense, but based on the average embedding of all discrete token embeddings after sequence tokenization*. Through

tokenization, two expressions that are very different in the mathematical sense (*e.g.*, Two numbers that are far apart on the real number line) may share many tokens because regular mathematical expressions are only taken from 0-9 number tokens and a limited number of special symbols, such as decimal points, slashes, and root signs. Thus, the collapse occurs at the token level during the autoregressive prediction of each token.

We conduct the following statistical analysis to demonstrate the universality of "*pattern collapse*" across various mathematical reasoning tasks: We categorize the mathematical tasks into various types across different domains and difficulties. In each task, we count the token number $N$ and the token type number $N_{\mathcal{T}}$ in the dataset, then compute the token duplication rate $D$ and the vocab coverage $C$ as follows: We use Llama-2 tokenizator [50], whose token type number in vocab $N_{\mathcal{V}}$ is 32000. The computation metric is:

$$D = 1 - \frac{N_{\mathcal{T}}}{N}, \quad C = \frac{N_{\mathcal{T}}}{N_{\mathcal{V}}}. \quad (8)$$

We also test translation and summarization tasks by taking samples with the same token size as the mathematical reasoning dataset for a clear comparison. Table 6 presents the statistics data, we find that: (i) The average token duplication rate was 98.9% on all math tasks, and even a staggering 99.9% on some easy arithmetic tasks; In contrast, the token duplication rate on the text

Table 6: Statistics about output tokens on mathematical reasoning (seven different task types) and text generation (translation and summarization).

| Task Type | $N$ | $N_{\mathcal{T}}$ | $D$ | $C$ |
|---|---|---|---|---|
| Mathematical Reasoning | | | | |
| Arithmetic(easy) | 16136 | 14 | 99.9% | 0.04% |
| Arithmetic(hard) | 5663 | 16 | 99.7% | 0.05% |
| Algebra | 5234 | 107 | 98.0% | 0.33% |
| Geometry | 2615 | 75 | 97.1% | 0.23% |
| Cnt.&Prob. | 2524 | 43 | 98.3% | 0.13% |
| Num.Theory | 2395 | 71 | 97.1% | 0.22% |
| Precalculus | 3388 | 84 | 97.5% | 0.26% |
| *Average* | *5422* | *58* | *98.9%* | *0.18%* |
| Text Generation | | | | |
| Translation | 2500 | 1065 | 57.4% | 3.32% |
| | 5000 | 1832 | 63.3% | 5.10% |
| | 10000 | 2980 | 70.2% | 9.31% |
| *Average* | *5833* | *1959* | *66.4%* | *6.12%* |
| Summarization | 2500 | 1265 | 49.4% | 4.01% |
| | 5000 | 1970 | 60.6% | 6.16% |
| | 10000 | 3192 | 68.0% | 9.98% |
| *Average* | *5833* | *2142* | *63.2%* | *6.69%* |

generation task is only about 60%, with about 2000 token types, and still increasing as the total number of tokens increases. These data and comparisons demonstrate that pattern collapse occurs on mathematical reasoning and not on text generation. (ii) Token repetition rate exceeded 97% on all seven math tasks of different difficulties and types. From these conclusions, we can demonstrate that the "*pattern collapse*" occurs on generally all types of mathematical reasoning tasks.

### 6.3 Can Chain-of-Thought Address "*pattern collapse*"?

A straightforward approach to addressing "*pattern collapse*" in the output space is to leverage chain-of-thought (CoT) techniques [54, 17, 62, 52, 61] to **expand the output space size**. Likewise, we adopt the solution steps associated with each sample in the MATH dataset as the output and employ SimCSE to derive embedding representations. The experimental setup aligns with Section 4.1, and the results are shown in Table 5(b). We note a similar phenomenon as in Table 5(a), *i.e.*, the detection accuracy under different ID-OOD pairs varies greatly, and thus the detection randomness is more pronounced. This suggests that despite that CoT expands the output space size, the output answer is still essentially related to the difficulty and digit of mathematical reasoning, and the semantic embedding representation cannot reflect these features accurately.

## 7 Conclusion

We propose the TV Score, a lightweight OOD detection method for mathematical reasoning, which distinguishes between ID and OOD samples by the embedding trajectory volatility in the latent space. We identify bottlenecks in OOD detection for mathematical reasoning and prove them empirically and theoretically. Experiments show that our method substantially outperforms all traditional algorithms, and can be extended to more application scenarios beyond mathematical reasoning.

## Limitations

Our limitation mainly lies in the relatively small sizes of datasets used in our experiments. Due to the difficulty of collecting and labeling mathematical reasoning data, dataset sizes in this field are generally small, mostly in the hundreds or thousands, making it difficult to obtain millions of data for training and reasoning as in translation or summarization tasks. To address this, we adopt test sampling to reduce the randomness under small-scale testing and mitigate the data imbalance.

## Ethics Statement

The data and models used in this work are sourced from the official version of the original paper, and we strictly adhere to the provided usage protocol. Regarding the data, no modifications have been made to the original dataset. Regarding the models, supervised fine-tuning and OOD data inference are involved. To mitigate the risk of uncontrollable outputs, all generated outputs in the experiments have been reviewed to ensure their safety. Furthermore, as our focus is solely on mathematical reasoning and does not involve sensitive content, we would not cause any potential societal impact.

## Acknowledgment

This work is supported by the Alibaba Innovative Research Program. Yiming and Rui are partially supported by the General Program of the National Natural Science Foundation of China (62176153), the Shanghai Municipal Science and Technology Major Project (2021SHZDZX0102), and SMP-Zhipu.AI Large Model Cross-Disciplinary Fund, the Science and Technology Development Fund. Zhuosheng is partially supported by the National Natural Science Foundation of China (62406188). This work is also partially supported by the Science and Technology Development Fund, Macau SAR (Grant Nos. FDCT/060/2022/AFJ, FDCT/0070/2022/AMJ), and the Multi-year Research Grant from the University of Macau (Grant No. MYRG-GRG2023-00006-FST-UMDF).

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

# Appendix

## A  Related Work

**Data-unavailable OOD Detection: Generic Methods.**    In scenarios where OOD data is unavailable. Detection methods are categorized into three main types: (1) Output-based methods assess confidence using predicted probabilities. (2) Ensemble-based methods assess the uncertainty of a collection of supporting models, classical techniques are Monte-Carlo dropout [46, 8] based on Bayesian inference and deep ensemble [19, 31]; (3) Feature-based methods assess the Mahalanobis Distance between OOD samples and the distribution of the ID data feature space, usually considering the input and output spaces [22, 42, 48], occasionally extending to specific hidden layer spaces [1]. Besides, there are some fragmented methods, such as gradient methods [23, 15] and autoencoder reconstruction methods [7, 16]. Still, these methods suffer from optimization and computational complexity with serious performance bottlenecks [58] and thus are not mainstream detection methods.

**OOD Detection in GLMs.**    Relatively few studies have explored OOD detection in GLMs, mainly in semantic parsing [31, 25], speech recognition [33], machine translation [56, 32], summarization [43], and they do not jump out of the frameworks of uncertainty estimation, ensemble-based methods, and embedding-based methods. To our knowledge, **we are the first to study OOD detection on mathematical reasoning, and we have found the failure of traditional algorithms in this scenario.** Mathematical reasoning is an important and difficult research topic in the era of LLMs, and this research is valuable for the scenario expansion of OOD detection algorithms on language models.

## B  Motivation Visualization Details of Figure 1

### B.1  Projection Setting

We use UMAP [34] for projection visualization, which is a nonlinear dimensionality reduction technique for mapping high-dimensional data into low-dimensional spaces. It uses optimization of the local and global structure of the original data to produce a high-quality mapping that can preserve the original data's local and global characteristics.

The UMAP algorithm consists of two steps: (i) Calculate the local similarity between each data point and its immediate neighbors to construct a local similarity map. (ii) Map the high-dimensional data to a low-dimensional space by optimizing an objective function that maintains the local and global structure. The hyperparameter "`n_neighbors`" in the first step is key, we set it as 10 in our paper.

### B.2  More Examples

We select the MATH [6] dataset for mathematical reasoning and the OPUS [49] for text generation (translation). In the MATH, we choose four domains: algebra, geometry, number theory, and precalculus; In the OPUS, we also choose four domains: Bible, news, TED, and health. We present examples of inputs and outputs under different domains for mathematical reasoning and translation scenarios in Table 7 and Table 8, respectively. They correspond to the case projection of Figure 1. Obviously, under mathematical reasoning, the outputs under different domains may appear exactly the same, while it is impossible under translation.

## C  Theoretical Analysis of Section 2.1

In this section, we theoretically analyze *Why pattern collapse in the output space leads to a greater likelihood of volatility differences in trajectories under different samples,* which corresponds to Hypothesis 1 in the main paper.

### C.1  Problem Setup

**Latent Space Embedding.**    For a given sample $s_i$, $[\boldsymbol{y}_i]_l \in \mathbb{R}^d (1 \leq l \leq L)$ denote the hidden embedding of the $l$-th layer. We define the embedding coordinate in the latent space as $\{(l, [\boldsymbol{y}_i]_l)\}_{l=1}^{L}$.

Table 7: Examples of inputs/outputs from different domains in the mathematical reasoning scenario.

| Input | Output |
|---|---|
| *Domain 1: Algebra* | |
| Suppose that $f$ is a function and $f^{-1}$ is the inverse of $f$. If $f(1) = 2$, $f(2) = 6$, and $f(3) = 5$, then what is $f^{-1}(f^{-1}(6))$? | 1 |
| *Domain 2: Geometry* | |
| Two chords, $AB$ and $CD$, meet inside a circle at $P$. If $AP = CP = 7$, then what is $\frac{BP}{DP}$? | 1 |
| *Domain 3: Number Theory* | |
| The product of the proper positive integer factors of $n$ can be written as $n^{(ax+b)/c}$, where $x$ is the number of positive divisors $n$ has, $c$ is a positive integer, and the greatest common factor of the three integers $a$, $b$, and $c$ is 1. What is $a + b + c$? | 1 |
| *Domain 4: Precalculus* | |
| Simplify $$\frac{1}{1 - \tan^2 x} + \frac{1}{1 - \cot^2 x}.$$ | 1 |

Table 8: Examples of inputs/outputs from different domains in the translation scenario.

| Input | Output |
|---|---|
| *Domain 1: Bible* | |
| The earth yielded grass, herbs yielding seed after their kind, and trees bearing fruit, with its seed in it, after their kind; and God saw that it was good. | 于是地发生了青草，和结种子的菜蔬，各从其类；并结果子的树木，各从其类；果子都包着核。神看着是好的。 |
| *Domain 2: News* | |
| Supporters say the tunnels would benefit the environment and offer Californians a more secure water supply. | 支持者们表示，这两条隧道将使环境受益并帮助加利福尼亚州确保供水更加安全。 |
| *Domain 3: TED* | |
| (Applause) June Cohen: Frank, that was beautiful, so touching. | （鼓掌） 主持：Frank 刚刚真是太美丽、太感人啦！ |
| *Domain 4: Health* | |
| The White House Coronavirus Task Force was established on January 29. | 白宫冠状病毒工作组（White House Coronavirus Task Force）是成立于2020年1月29日的以应对美国COVID-19疫情的工作组。 |

**Embedding Interpolation.** We assume the $\boldsymbol{F}_i(x) = [F_{i1}(x), F_{i2}(x), ..., F_{id}(x)]^\top : \mathbb{R} \to \mathbb{R}^d$ with $d$ independent components fits the $L$ coordinates, representing a continuous learning trajectory for $s_i$, so $[\boldsymbol{y}_i]_l = \boldsymbol{F}_i(l)$. $\boldsymbol{F}_i(x)$ is taken from the *functional space* $\boldsymbol{X}$, so $\{\boldsymbol{F}_i(l)\}_{l=1}^L$ are all independent variables. We constrain each component function of this $\boldsymbol{F}_i(x)$ to satisfy the $m$-order ($m \geq 3$) derivability property. Under this setting, the definition of dimension-independent trajectory volatility (Eq. 2) equates to

$$
\begin{aligned}
\boldsymbol{V}(s_i) &= [V_1(s_i), V_2(s_i), ..., V_L(s_i)]^\top \\
&= \frac{1}{L} \sum_{l=1}^{L-1} [|F_{i1}(l) - F_{i1}(l-1)|, |F_{i2}(l) - F_{i2}(l-1)|, ..., |F_{id}(l) - F_{id}(l-1)|]^\top
\end{aligned}
\tag{9}
$$

**Modeling.** We observe the "*pattern collapse*" phenomenon in the output space in Figure 1. We abstract this phenomenon in Figure 2, which compares the trajectory trend between different samples in the mathematical reasoning and text generation scenarios.

We specify that $[\boldsymbol{y}_i]_L = \boldsymbol{F}_i(L) \sim \mathcal{N}(\boldsymbol{c}, \Sigma^2)$, where

$$\boldsymbol{c} = [c_1, c_2, ..., c_d]^\top, \quad \Sigma = \mathrm{diag}(\delta_1, \delta_2, \cdots, \delta_d). \tag{10}$$

According to the pattern collapse property under different tasks, we can constraint the endpoint embedding $\boldsymbol{F}_i(L)$ in the output space:

- For mathematical reasoning with pattern collapse, $\Sigma \to \boldsymbol{O}$, so we approximate that $\boldsymbol{F}_i(L) \equiv \boldsymbol{c}$;
- For text generation without pattern collapse, $\Sigma \neq \boldsymbol{O}$, so $\boldsymbol{F}_i(L) \not\equiv \boldsymbol{c}$.

With such constraints, we model the main theorem:

**Theorem C.1 (Main Theorem)** *For different samples $s_i$ and $s_j$, the likelihood of variations in trajectory volatility under mathematical reasoning scenarios is higher than that under text generation scenarios, which means:*

$$\mathbb{E}_{\{\boldsymbol{F}_i(l)\}_{l=1}^L, \, \{\boldsymbol{F}_j(l)\}_{l=1}^L \sim \mathcal{U}(\mathbb{R}^d)} \left\{ \boldsymbol{V}(s_i) - \boldsymbol{V}(s_j) \neq \boldsymbol{0} | \boldsymbol{F}_i(L), \boldsymbol{F}_j(L) \equiv \boldsymbol{c} \right\}$$
$$> \mathbb{E}_{\{\boldsymbol{F}_i(l)\}_{l=1}^L, \, \{\boldsymbol{F}_j(l)\}_{l=1}^L \sim \mathcal{U}(\mathbb{R}^d)} \left\{ \boldsymbol{V}(s_i) - \boldsymbol{V}(s_j) \neq \boldsymbol{0} | \boldsymbol{F}_i(L), \boldsymbol{F}_j(L) \sim \mathcal{N}(\boldsymbol{c}, \Sigma^2) \right\},$$

*where $\mathcal{U}(\mathbb{R}^d)$ denotes the uniform distribution defined in $d$-dimensional real number space.*

## C.2 Prelinimary

Next, we move on to the formal proofs. We begin with some propositions and lemmas that will be useful in the main theorem.

**Proposition C.1 (Lagrange Remainder Term)** *In the Taylor Expansion expression*

$$f(x) = f(a) + \frac{\mathrm{d}f}{\mathrm{d}x}(x-a) + \frac{1}{2!} \cdot \frac{\mathrm{d}^2 f}{\mathrm{d}x^2}(x-a)^2 + \cdots + \frac{1}{n!} \cdot \frac{\mathrm{d}^n f}{\mathrm{d}x^n}(x-a)^n + R_{n+1}(x),$$

*the remainder $R_{n+1}(x)$ has the following property:*

$$|R_{n+1}(x)| = \left| \frac{1}{(n+1)!} \cdot \frac{\mathrm{d}^{n+1} f}{\mathrm{d}c^{n+1}} \cdot (x-a)^{n+1} \right| \leq \frac{M}{(n+1)!} \cdot \left| (x-a)^{n+1} \right|,$$

*where $c \in [a, x]$ or $c \in [x, a]$, and $M = \sup \left\{ \left| \frac{\mathrm{d}^{n+1} f}{\mathrm{d}\xi^{n+1}} \right| : \xi \in [a, x] \text{ or } \xi \in [x, a] \right\} > 0.$*

*Proof.* We consider the case of $a < x$ with $a > x$ identically. We use the Fundamental Theorem of Calculus (FTC) for the most basic expansion of $f(x)$:

$$f(x) = f(a) + \int_a^x \frac{\mathrm{d}f}{\mathrm{d}x_1} \, \mathrm{d}x_1. \tag{11}$$

Continue to use the FTC to expand the derivatives in integrals:

$$f(x) = f(a) + \int_a^x \frac{\mathrm{d}f}{\mathrm{d}x_1} \, \mathrm{d}x_1 = f(a) + \int_a^x \left( \frac{\mathrm{d}f}{\mathrm{d}a} + \int_a^{x_1} \frac{\mathrm{d}^2 f}{\mathrm{d}x_2^2} \, \mathrm{d}x_2 \right) \mathrm{d}x_1$$
$$= f(a) + \frac{\mathrm{d}f}{\mathrm{d}x}(x-a) + \int_a^x \int_a^{x_1} \frac{\mathrm{d}^2 f}{\mathrm{d}x_2^2} \, \mathrm{d}x_2 \, \mathrm{d}x_1$$
$$= \cdots$$
$$= \sum_{k=0}^n \frac{1}{k!} \cdot \frac{\mathrm{d}^k f}{\mathrm{d}a^k} \cdot (x-a)^k + \int_a^x \int_a^{x_1} \int_a^{x_2} \cdots \int_a^{x_n} \frac{\mathrm{d}^{n+1} f}{\mathrm{d}x_{n+1}^{n+1}} \, \mathrm{d}x_{n+1} \, \mathrm{d}x_n \cdots \mathrm{d}x_1. \tag{12}$$

Therefore, the generalized remainder is known as

$$R_{n+1}(x) = \int_a^x \int_a^{x_1} \int_a^{x_2} \cdots \int_a^{x_n} \frac{\mathrm{d}^{n+1} f}{\mathrm{d}x_{n+1}^{n+1}} \, \mathrm{d}x_{n+1} \, \mathrm{d}x_n \cdots \mathrm{d}x_1$$
$$= \int_a^x \frac{1}{n!} \cdot \frac{\mathrm{d}^{n+1} f}{\mathrm{d}t^{n+1}} \cdot (x-t)^n \, \mathrm{d}t. \tag{13}$$

We let $m_{n+1} = \min_{t\in[a,x]} \frac{\mathrm{d}^{n+1}f}{\mathrm{d}t^{n+1}}$ and $M_{n+1} = \max_{t\in[a,x]} \frac{\mathrm{d}^{n+1}f}{\mathrm{d}t^{n+1}}$, so

$$m_{n+1}\int_a^x (x-t)^n\,\mathrm{d}t \le \int_a^x \frac{\mathrm{d}^{n+1}f}{\mathrm{d}t^{n+1}}\cdot(x-t)^n\,\mathrm{d}t \le M_{n+1}\int_a^x (x-t)^n\,\mathrm{d}t$$

$$\implies m_{n+1} \le \frac{\int_a^x \frac{\mathrm{d}^{n+1}f}{\mathrm{d}t^{n+1}}\cdot(x-t)^n}{\frac{(x-a)^{n+1}}{n+1}} \le M_{n+1}. \tag{14}$$

According to the Lagrange's Mean Value Theorem, there must exist a number $c \in [a,x]$ with

$$\frac{\mathrm{d}^{n+1}f}{\mathrm{d}c^{n+1}} = \frac{\int_a^x \frac{\mathrm{d}^{n+1}f}{\mathrm{d}t^{n+1}}\cdot(x-t)^n}{\frac{(x-a)^{n+1}}{n+1}}, \tag{15}$$

this gives us

$$\int_a^x \frac{\mathrm{d}^{n+1}f}{\mathrm{d}t^{n+1}}\cdot(x-t)^n = \frac{\mathrm{d}^{n+1}f}{\mathrm{d}c^{n+1}}\cdot\frac{(x-a)^{n+1}}{n+1}$$

$$\implies R_{n+1}(x) = \frac{1}{(n+1)!}\cdot\frac{\mathrm{d}^{n+1}f}{\mathrm{d}c^{n+1}}\cdot(x-a)^{n+1}, \tag{16}$$

and the equation on the left side of the proposition is proved completely. The inequality on the right-hand side is clearly established by the Lagrange's Mean Value Theorem. $\square$

**Lemma C.1 (Error Bound for the Midpoint Rule)** *Suppose that $f(x)$ is a $m$-th ($m \ge 2$) order differentiable function on the interval $(-\infty, +\infty)$, and $K = \sup\left\{\left|\frac{\mathrm{d}^2 f}{\mathrm{d}x^2}\right| : x \in [a,b]\right\} \in \mathbb{R}$, then*

$$\left|\int_a^b f(x)\,\mathrm{d}x - (b-a)f\left(\frac{a+b}{2}\right)\right| \le \frac{K}{24}(b-a)^3$$

*Proof.* We do the first order Taylor Expansion for $f(x)$ at the midpoint $x = \frac{a+b}{2}$ of the interval $[a,b]$:

$$\left|\int_a^b f(x)\,\mathrm{d}x - (b-a)f\left(\frac{a+b}{2}\right)\right| = \left|\int_a^b \left[f(x) - f\left(\frac{a+b}{2}\right)\right]\mathrm{d}x\right|$$

$$= \left|\int_a^b \left[\frac{\mathrm{d}f}{\mathrm{d}\left(\frac{a+b}{2}\right)}\cdot\left(x - \frac{a+b}{2}\right) + R_1(x)\right]\mathrm{d}x\right| \tag{17}$$

$$= \left|0 + \int_a^b R_1(x)\,\mathrm{d}x\right| = \left|\int_a^b R_1(x)\,\mathrm{d}x\right|$$

Following Theorem C.1, we do the inequality scaling as below:

$$\left|\int_a^b R_1(x)\,\mathrm{d}x\right| \le \left|\int_a^b \frac{K}{2!}\left(x - \frac{a+b}{2}\right)^2 \mathrm{d}x\right| = \frac{K}{6}\left[\left(\frac{b-a}{2}\right)^3 - \left(\frac{a-b}{2}\right)^3\right] = \frac{K}{24}(b-a)^3 \tag{18}$$

$\square$

**Lemma C.2 (Differential-Integral Error Order Estimation)** *Suppose that $f(x)$ is a $m$-th ($m \ge 3$) order differentiable function on the interval $(-\infty, +\infty)$ and $K_i = \sup\left\{\left|\frac{\mathrm{d}^i f}{\mathrm{d}x^i}\right| : x \in (-\infty, +\infty)\right\} < +\infty$ ($i = 1, 2, \cdots, m$), then on the closed interval $[a,b]$, we have*

$$\mathcal{A} := \left\{\left|\sum_{i=1}^L \left|f\left(a + \frac{b-a}{L}i\right) - f\left(a + \frac{b-a}{L}(i-1)\right)\right| - \int_a^b \left|\frac{\mathrm{d}f}{\mathrm{d}x}\right|\mathrm{d}x\right|\right\} \le R(L) \sim o(\frac{1}{L}).$$

*Proof.* For each sub-interval $\left(a + \frac{b-a}{L}(i-1), a + \frac{b-a}{L}i\right)$, we perform a Taylor Expansion at the midpoint $x = a + \frac{b-a}{L}(i - \frac{1}{2})$ to obtain the following expression:

$$
\sum_{i=1}^{L} \left| f(a + \frac{b-a}{L}i) - f(a + \frac{b-a}{L}(i-1)) \right|
$$

$$
= \sum_{i=1}^{L} \left| \left[ f\left(a + \frac{b-a}{L}(i - \frac{1}{2})\right) + \frac{\mathrm{d}f}{\mathrm{d}\left(a + \frac{b-a}{2L}(i - \frac{1}{2})\right)} \cdot \frac{b-a}{L} + \mathcal{O}\left[\left(\frac{b-a}{2L}\right)^2\right] \right] \right.
$$

$$
\left. - \left[ f\left(a + \frac{b-a}{L}(i - \frac{1}{2})\right) + \frac{\mathrm{d}f}{\mathrm{d}\left(a + \frac{b-a}{2L}(i - \frac{1}{2})\right)} \cdot \frac{a-b}{L} + \mathcal{O}\left[\left(\frac{b-a}{2L}\right)^2\right] \right] \right|
\tag{19}
$$

$$
= \sum_{i=1}^{L} \left| \frac{\mathrm{d}f}{\mathrm{d}\left(a + \frac{b-a}{L}(i - \frac{1}{2})\right)} \cdot \frac{b-a}{L} + \mathcal{O}(\frac{1}{L^2}) \right|
$$

$$
\leq \frac{b-a}{L} \cdot \sum_{i=1}^{L} \left| \frac{\mathrm{d}f}{\mathrm{d}\left(a + \frac{b-a}{L}(i - \frac{1}{2})\right)} \right| + \sum_{i=1}^{L} \mathcal{O}(\frac{1}{L^2})
$$

$$
= \frac{b-a}{L} \cdot \sum_{i=1}^{L} \left| \frac{\mathrm{d}f}{\mathrm{d}\left(a + \frac{b-a}{L}(i - \frac{1}{2})\right)} \right| + \mathcal{O}(\frac{1}{L})
$$

Following Lemma C.1, we do the inequality scaling for the first-order derivative summation:

$$
\frac{b-a}{L} \cdot \sum_{i=1}^{L} \left| \frac{\mathrm{d}f}{\mathrm{d}\left(a + \frac{b-a}{L}(i - \frac{1}{2})\right)} \right| \leq \sum_{i=1}^{L} \int_{a + \frac{b-a}{L}(i-1)}^{a + \frac{b-a}{L}i} \left[ \left| \frac{\mathrm{d}f}{\mathrm{d}x} \right| + \frac{K_3}{24} \left(\frac{b-a}{L}\right)^3 \right] \mathrm{d}x
$$

$$
= \int_a^b \left[ \left| \frac{\mathrm{d}f}{\mathrm{d}x} \right| + \frac{K_3(b-a)^3}{24L^2} \right] \mathrm{d}x = \int_a^b \left| \frac{\mathrm{d}f}{\mathrm{d}x} \right| \mathrm{d}x + \frac{K_3(b-a)^4}{24L^2}
\tag{20}
$$

$$
\mathcal{A} :\leq \frac{K_3(b-a)^4}{24L^2} + \mathcal{O}(\frac{1}{L}) = R(L) \sim o(\frac{1}{L})
\tag{21}
$$

$\square$

### C.3 Main Theorem Proof

Finally, we formally prove our main theorem C.1 in this sub-section, thereby verifying the Hypotheses 1 in the main paper. Since the $d$ components of $\boldsymbol{V}(s_i) - \boldsymbol{V}(s_j)$ are independent of each other, we only consider the $d$-th dimensional component and the rest of the components $V_d(s_i) - V_d(s_j)$ can be proved in the same way.

*Proof.* According to Lemma C.2, we can approximate $V_d(s_i) - V_d(s_j)$ as

$$
V_d(s_i) - V_d(s_j) = \sum_{l=1}^{L} |F_{id}(l) - F_{id}(l-1)| - \sum_{l=1}^{L} |F_{jd}(l) - F_{jd}(l-1)|
$$

$$
= \int_1^L \left| \frac{\mathrm{d}F_{id}}{\mathrm{d}x} \right| \mathrm{d}x - \int_1^L \left| \frac{\mathrm{d}F_{jd}}{\mathrm{d}x} \right| \mathrm{d}x + \mathcal{O}(\frac{1}{L})
\tag{22}
$$

$$
\approx \int_1^L \left| \frac{\mathrm{d}F_{id}}{\mathrm{d}x} \right| \mathrm{d}x - \int_1^L \left| \frac{\mathrm{d}F_{jd}}{\mathrm{d}x} \right| \mathrm{d}x
$$

Now we want to remove the absolute value of the integrated function. Due to the continuity of the first-order derivative, there must be several zero points that are not extreme points, dividing the domain of function definition into several open intervals with alternating constant positive and negative function values. We first define such a set of zeros on the domain of definition $\mathcal{D} = [1, L]$:

$$
\mathcal{X}_i = \left\{ x \mid x \in \mathcal{D}; \frac{\mathrm{d}F_{id}}{\mathrm{d}x} = 0 \right\} = \{i_1, i_2, \cdots, i_p\},
$$

$$
\mathcal{X}_j = \left\{ x \mid x \in \mathcal{D}; \frac{\mathrm{d}F_{jd}}{\mathrm{d}x} = 0 \right\} = \{j_1, j_2, \cdots, j_q\}.
\tag{23}
$$

For all zero points $i_t \in \mathcal{X}_i$ or $j_t \in \mathcal{X}_j$, they must satisfy the following properties to ensure that they are not extreme points:

$$\exists \epsilon > 0, \forall \Delta x < \epsilon, \frac{\mathrm{d}F_{id}}{\mathrm{d}(i_t - \Delta x)} \cdot \frac{\mathrm{d}F_{id}}{\mathrm{d}(i_t + \Delta x)} < 0, \tag{24}$$

and $j_t$ is the same as $i_t$. Therefore, in the sub-interval $(1, i_1), (i_1, i_2), \cdots, (i_p, L)$, $F_{id}(x)$ is alternately constant positive and constant negative, and the same as $F_{jd}(x)$.

We assume that the first interval $(1, i_1)$ and $(1, j_1)$ are constant positive intervals (the same can be proven for constant negative intervals). In this setting, we can continue to simplify Eq.22:

$$
\begin{aligned}
&\int_1^L \left| \frac{\mathrm{d}F_{id}}{\mathrm{d}x} \right| \mathrm{d}x - \int_1^L \left| \frac{\mathrm{d}F_{jd}}{\mathrm{d}x} \right| \mathrm{d}x \\
&= \left( \int_1^{i_1} \left| \frac{\mathrm{d}F_{id}}{\mathrm{d}x} \right| \mathrm{d}x + \int_{i_1}^{i_2} \left| \frac{\mathrm{d}F_{id}}{\mathrm{d}x} \right| \mathrm{d}x + \cdots + \int_{i_p}^L \left| \frac{\mathrm{d}F_{id}}{\mathrm{d}x} \right| \mathrm{d}x \right) \\
&\quad - \left( \int_1^{j_1} \left| \frac{\mathrm{d}F_{jd}}{\mathrm{d}x} \right| \mathrm{d}x + \int_{j_1}^{j_2} \left| \frac{\mathrm{d}F_{jd}}{\mathrm{d}x} \right| \mathrm{d}x + \cdots + \int_{j_q}^L \left| \frac{\mathrm{d}F_{jd}}{\mathrm{d}x} \right| \mathrm{d}x \right) \\
&= \left[ F_{id}(x)|_1^{i_1} + (-1) \cdot F_{id}(x)|_{i_1}^{i_2} + \cdots + (-1)^{i_p} \cdot F_{id}(x)|_{i_p}^L \right] \\
&\quad - \left[ F_{jd}(x)|_1^{j_1} + (-1) \cdot F_{jd}(x)|_{j_1}^{j_2} + \cdots + (-1)^{j_q} \cdot F_{jd}(x)|_{j_q}^L \right] \\
&= [-F_{id}(1) + F_{jd}(1)] + \left[ 2 \sum_{k=1}^p (-1)^{k-1} \cdot F_{id}(i_k) - 2 \sum_{k=1}^q (-1)^{k-1} \cdot F_{jd}(j_k) \right] \\
&\quad + \left[ (-1)^{i_p - 1} \cdot F_{id}(L) - (-1)^{j_q - 1} \cdot F_{jd}(L) \right]
\end{aligned}
\tag{25}
$$

Since $F_{id}(x)$ and $F_{jd}(x)$ are taken from the functional space, $\{F_{id}(1), \{F_{id}(i_k)\}_{k=1}^p\}$ and $\{F_{jd}(1), \{F_{jd}(j_k)\}_{k=1}^q\}$ can be seen as independent variables. We let

$$c = \left[ (-1)^{i_p - 1} \cdot F_{id}(L) - (-1)^{j_q - 1} \cdot F_{jd}(L) \right], \tag{26}$$

then we can rewrite Eq.25 to the matrix form:

$$\int_1^L \left| \frac{\mathrm{d}F_{id}}{\mathrm{d}x} \right| \mathrm{d}x - \int_1^L \left| \frac{\mathrm{d}F_{jd}}{\mathrm{d}x} \right| \mathrm{d}x = \boldsymbol{A}\boldsymbol{x} + \boldsymbol{c}, \tag{27}$$

where $\boldsymbol{A}$ is the coefficient matrix, and $\boldsymbol{x}$ is the unknown variable vector:

$$\boldsymbol{A} = [-1, 1, \underbrace{2, -2, 2, -2, ..., (-1)^{p-1} \cdot 2}_{p \text{ dimensions}}, \underbrace{-2, 2, -2, 2, ..., (-1)^q \cdot 2}_{q \text{ dimensions}}] \in \mathbb{R}^{1 \times (p+q+2)}$$

$$\boldsymbol{x} = [F_{id}(1), F_{jd}(1), F_{id}(i_1), ..., F_{id}(i_p), F_{jd}(j_1), ..., F_{jd}(j_q)]^\top \in \mathbb{R}^{(p+q+2) \times 1}
\tag{28}$$

We let $i_p \equiv j_q \pmod 2$, and return to the condition in the mathematical expectation on both sides of the inequality in the main theorem C.1 for the following categorical discussion:

- If $F_{id}(L) = F_{jd}(L) = c_d$, then $\boldsymbol{c} = \boldsymbol{0}$, so $V_d(s_i) - V_d(s_j) = 0 \implies \boldsymbol{A}\boldsymbol{x} = \boldsymbol{0}$;
- If $F_{id}(L), F_{jd}(L) \sim \mathcal{N}(c_d, \delta_d^2)$, then $\boldsymbol{c} \not\equiv \boldsymbol{0}$, so $V_d(s_i) - V_d(s_j) \neq 0 \implies \boldsymbol{A}\boldsymbol{x} = -\boldsymbol{c}$.

We denote $N(A)$ as the solution space of $\{\boldsymbol{x} | \boldsymbol{A}\boldsymbol{x} = \boldsymbol{0}\}$ and $P(A)$ as the solution space of $\{\boldsymbol{x} | \boldsymbol{A}\boldsymbol{x} = -\boldsymbol{c}\}$. For $x \in P(A)$, $x = x_p$ or $x = x_p + x_n$ where $x_p$ is the special solution and $x_n$ is the solution in zero space, $i.e.$, $x_n \in N(A)$. Since $\mathrm{rank}(A) = 1 < p + q + 2$, the special solution $x_p$ exists, so the size of the solution space $P(A)$ is larger than $N(A)$. In this case, when the variable $\boldsymbol{x}$ is sampled from the real number space, its probability of being in $N(A)$ is smaller, namely its probability of not being in $N(A)$ is larger. This is exactly equivalent to the form of the proof of Main Theorem C.1, so the proof is complete. $\qquad \square$

## C.4 Extended Conclusion

During the proof of the main theorem, we can unlock a hidden conclusion due to the embedding interpolation: *GLMs with a larger number of hidden layers may achieve more stable detection performance,* i.e., *discrepancies in embedding volatility of different samples will be more obvious.*

In Lemma C.2, we have proved the upper bound of differential-integral error order estimation is the equivalent infinitesimals of $1/L$. Actually, when $L \to +\infty$, we have

$$\lim_{L \to +\infty} \sum_{i=1}^{L} \left| f\left(a + \frac{b-a}{L}i\right) - f\left(a + \frac{b-a}{L}(i-1)\right) \right| = \int_{a}^{b} \left| \frac{\mathrm{d}f}{\mathrm{d}x} \right| \, \mathrm{d}x, \tag{29}$$

which is clear according to the definition of Riemann Sum. This means that when the value of $L$ increases, the differential-integral approximation error will be reduced, so the conclusion of the main theorem will be more accurate. In our experiments, we use GPT2-XL (48 layers) and Llama2-7B (32 layers) as training backbones and find that the average performance of GPT2-XL is higher and more stable than that of Llama2-7B, while the number of layers of GPT2-XL is 1.5 times that of Llama2-7B. This further validates the correctness of our extended conclusions.

# D   Algorithm: TV Score Computation

The pseudo-code of our TV Score computation pipeline is shown in Algorithm 1.

---
**Algorithm 1** Trajectory Volatility (TV) Score Computation

---
**Input:** $L$: The number of hidden layers
    $N$: The size of ID dataset
    $k$: the smoothing differential order of TV score
    $\{\boldsymbol{y}_l\}_{1 \leq l \leq L}$: the average hidden embedding of the OOD sample in each layer
    $\{[\hat{\boldsymbol{y}}_l]_i\}_{1 \leq l \leq L, 1 \leq i \leq N}$: the average hidden embedding of all $N$ ID samples in each layer
1: **for** $l \leftarrow 1$ **to** $L$ **do**
2:     Fitting ID samples $\{[\hat{\boldsymbol{y}}_l]_i\}_{1 \leq l \leq L, 1 \leq i \leq N}$ to Gaussian distribution $\mathcal{G}_l = \mathcal{N}(\boldsymbol{\mu}_l, \boldsymbol{\Sigma}_l)$
3:     $f(\boldsymbol{y}_l) \leftarrow (\boldsymbol{y}_l - \boldsymbol{\mu}_l)^{\top} (\boldsymbol{\Sigma}_l)^{-1} (\boldsymbol{y}_l - \boldsymbol{\mu}_l)$
4: **end for**
5: **for** $i \to 1$ **to** $k$ **do**
6:     **for** $l \leftarrow 1$ **to** $L$ **do**
7:         $\Delta^{(i)}\mathcal{G}_l = \mathcal{N}(\boldsymbol{\mu}_l^{(i)}, \boldsymbol{\Sigma}_l^{(i)}) \leftarrow \mathcal{N}(\sum_{t=0}^{i}(-1)^{i+t}\mathrm{C}_i^t\boldsymbol{\mu}_{l+i}, \ \sum_{t=0}^{i}\mathrm{C}_i^t\boldsymbol{\Sigma}_{l+i})$
8:     **end for**
9:     $f^{(i)}(\boldsymbol{y}_l) \leftarrow \left(\boldsymbol{y}_l^{(i)} - \boldsymbol{\mu}_l^{(i)}\right)^{\top} \left(\boldsymbol{\Sigma}_l^{(i)}\right)^{-1} \left(\boldsymbol{y}_l^{(i)} - \boldsymbol{\mu}_l^{(i)}\right)$
10: **end for**
11: **for** $i \to 1$ **to** $n$ **do**
12:     $\mathrm{TV}_i \leftarrow$ `average` $\left[f^{(i)}(\boldsymbol{y}_l)\right]_{1 \leq l \leq L}$
13: **end for**
**Output:** $\{\mathrm{TV}_i\}_{1 \leq i \leq k}$

---

As for the computational complexity, once we have all $\boldsymbol{y}_l$, our TV score requires only scalar addition and multiplication. During the ID distribution fitting phase, both operations are $\mathcal{O}(Ldn)$, where $n$ is the dataset size. During the score computation phase, both operations are $\mathcal{O}(Ldk)$. In practice, the computation time is measured in milliseconds.

# E   Experimental Setting Details

## E.1   Basic Information of Dataset

For the ID dataset, we use the MultiArith [44], which consists of Math Word Problems on arithmetic reasoning. For the OOD datasets, we intuitively introduce two types of detection scenarios following [43]: **(i) Far-shift OOD** setting, we select the MATH [11] as the OOD data, which spans across

distinct mathematical domains encompassing algebra, geometry, counting and probability, number theory, and precalculus. It contains college difficulty level math problems, whereas MultiArith has only elementary school difficulty, and thus can be considered as sourced from far-different distributions; (ii) **Near-shift OOD** setting, we select five arithmetic reasoning datasets as the OOD data: GSM8K [6], SVAMP [39], AddSub [13], SingleEq [18], and SingleOp [18], they all consist of Math Word Problems like the MultiArith but require different reasoning hops and knowledge points for solving the problems, and thus can be considered as sourced from near-different distributions. In addition, we present the data sizes and examples of all ID and OOD datasets, as shown in Table 9.

### E.2 ID Dataset Split

For the ID dataset MultiArith, we find that every 100 consecutive samples show a similar quadratic operation pattern (*e.g.*, samples with id 0-100 are a mixture of addition and subtraction, and id 100-200 are a mixture of addition and multiplication). Therefore, we divide it into 6 subsets (6*100) in order. In each subset, we take the first 60 as training samples and the last 40 as test samples.

### E.3 Training Implementation

We train Llama2-7B [50] and GPT2-XL [5] models on the training split of MultiArith. Llama2-7B is trained with AdamW optimizer [29] for 10K steps and 8 batch sizes in 4-card RTX 3090 (2 per card). The learning rate is set to 1e-5, the warmup step to 10, and the maximum gradient normalization to 0.3. GPT2-XL is trained for 3K steps and 128 batch sizes in a single RTX 3090, and other configurations are the same as Llama2-7B.

### E.4 OOD Dataset Selection Rationality

In this part, we examine the rationality of the OOD data selection, ensuring that the OOD data distribution significantly differs from the pre-trained data distribution and has not been fully learned during the pre-training phase. Some research [57, 59] have confirmed the absence of data leakage in Llama2 for MATH and GSM8K datasets, we still conduct experiments and analyses to ensure this.

For GPT2-XL, We can determine this from the time dimension. GPT-2 was released in 2019, but the MATH and GSM8k datasets were released in 2021, so they are unlikely to appear in the pre-training data. However, for Llama2-7B, due to the closed-source data, we cannot confirm which data the model used in the pre-training phase, so we cannot fully determine whether the selected dataset was OOD for the model from a data perspective.

Therefore, we argue that a dataset can be considered OOD when it is beyond the capability of the base model as claimed by prior work [47, 27]. We test all ten datasets we select as the OOD data in the pre-trained Llama2-7B and GPT2-XL model, and Table 10 shows the results. We find that GPT2-XL cannot handle any of the ten mathematical reasoning tasks, and Llama2-7B performs with very low accuracy. Therefore, from a capability standpoint, we can ensure that these datasets are OOD for these two GLMs.

### E.5 Baseline

Let $x$ represent the input sequence and $y$ the output sequence, with lengths denoted as $n_x$ and $n_y$ respectively. In addition, we assume that $p(\cdot; \theta)$ represents the GLM parameterized by $\theta$ that has been trained in ID dataset $\mathcal{D}$, outputting a sequence of softmax probabilities. We compare some training-free baseline methods as below:

- Maximum Softmax Probability [12]:

$$\frac{1}{n_y} \cdot \sum_{i=1}^{n_y} p(y_i | y_{\prec i}, x; \theta). \tag{30}$$

- Monte-Carlo Dropout [8]:

$$\int p(y|x; \theta) q(\theta) \mathrm{d}\theta, \tag{31}$$

where $q(\theta) \approx p(\theta | \mathcal{D})$.

Table 9: ID and OOD datasets used in this paper.

**In-Distribution Dataset**

**MultiArith (Data Size: 600)**

Q: Kaleb was collecting cans for recycling. On Saturday he filled 5 bags up and on Sunday he filled 5 more bags. If each bag had 4 cans in it, how many cans did he pick up total?

A: 40.0

| **Far Shift Out-of-Distribution Dataset** | **Near Shift Out-of-Distribution Dataset** |
| --- | --- |
| **MATH-Algebra (Data Size: 1187)** | **GSM8K (Data Size: 1318)** |
| Q: How many real numbers are not in the domain of the function $$f(x) = \frac{1}{x-64} + \frac{1}{x^2-64} + \frac{1}{x^3-64} ?$$ | Q: Judy teaches 5 dance classes, every day, on the weekdays and 8 classes on Saturday. If each class has 15 students and she charges \$15.00 per student, how much money does she make in 1 week? |
| A: 4 | A: 7425 |
| **MATH-Geometry (Data Size: 479)** | **SVAMP (Data Size: 1000)** |
| Q: Suppose we are given seven points that are equally spaced around a circle. If $P$, $Q$, and $R$ are chosen to be any three of these points, then how many different possible values are there for $m\angle PQR$? | Q: A mailman has to give 38 pieces of junk mail to each of the 78 blocks. If there are 19 houses on a block. How many pieces of junk mail should he give each house? |
| A: 5 | A: 2.0 |
| **MATH-Counting and Probability (Data Size: 474)** | **AddSub (Data Size: 395)** |
| Q: Amy's grandmother gave her 3 identical chocolate chip cookies and 4 identical sugar cookies. In how many different orders can Amy eat the cookies such that either she eats a chocolate chip cookie first, she eats a chocolate chip cookie last, or both? | Q: While taking inventory at her pastry shop, Kelly realizes that she had 0.4 box of baking powder yesterday, but the supply is now down to 0.3 box. How much more baking powder did Kelly have yesterday? |
| A: 25 | A: 0.1 |
| **MATH-Number Theory (Data Size: 540)** | **SingleEq (Data Size: 508)** |
| Q: Notice that $$31 \cdot 37 = 1147.$$ Find some integer $n$ with $0 \le n < 2293$ such that $$31n \equiv 3 \pmod{2293}.$$ | Q: Fred had 7 dimes in his bank. His sister borrowed 3 of his dimes. How many dimes does Fred have now? |
| A: 222 | A: 4.0 |
| **MATH-Precalculus (Data Size: 546)** | **SingleOp (Data Size: 562)** |
| Q: Let $\mathbf{a} = \begin{pmatrix} 1 \\ 3 \\ 0 \end{pmatrix}$, $\mathbf{b} = \begin{pmatrix} -11 \\ 5 \\ 2 \end{pmatrix}$, and $\mathbf{c} = \begin{pmatrix} 1+\sqrt{5} \\ 4 \\ -5 \end{pmatrix}$. Find $k$ if the vectors $\mathbf{a} + \mathbf{b} + \mathbf{c}$ and $$3(\mathbf{b} \times \mathbf{c}) - 8(\mathbf{c} \times \mathbf{a}) + k(\mathbf{a} \times \mathbf{b})$$ are orthogonal. | Q: Pamela starts with 30 bottle caps. Jean takes 26 away. How many bottle caps does Pamela end with? A: 4.0 |
| A: 5 | |

Table 10: Accuracies of all datasets we select as the OOD data in pre-trained GLMs.

| | Far-shift OOD Setting | | | Near-shift OOD Setting | |
| *Dataset* | **Llama2-7B** | **GPT2-XL** | *Dataset* | **Llama2-7B** | **GPT2-XL** |
| | Accuracy of pre-trained model | | | Accuracy of pre-trained model | |
| Algebra | 6 / 1187 | 0 / 1187 | GSM8K | 0 / 1318 | 0 / 1318 |
| Geometry | 2 / 479 | 0 / 479 | SVAMP | 8 / 1000 | 0 / 1000 |
| Cnt.&Prob | 4 / 474 | 0 / 474 | AddSub | 13 / 395 | 0 / 1000 |
| Num.Theory | 0 / 540 | 0 / 540 | SingleEq | 5 / 395 | 0 / 508 |
| Precalculus | 1 / 546 | 0 / 540 | SingleOp | 7 / 508 | 0 / 508 |

- Sequence Perplexity [43]:

$$\left[\prod_{i=1}^{n_y} p(y_i|y_{\prec i}, \boldsymbol{x}; \boldsymbol{\theta})\right]^{-\frac{1}{n_y}}. \tag{32}$$

- Input Embedding [43]:

$$(\boldsymbol{x} - \boldsymbol{\mu}_x)^\top \boldsymbol{\Sigma}_x^{-1} (\boldsymbol{x} - \boldsymbol{\mu}_x), \tag{33}$$

where $\boldsymbol{\mu}_x$ and $\boldsymbol{\Sigma}_x$ represent the mean and variance of the Gaussian distribution associated with the first-layer hidden state.

- Output Embedding [43]:

$$(\boldsymbol{y} - \boldsymbol{\mu}_y)^\top \boldsymbol{\Sigma}_y^{-1} (\boldsymbol{y} - \boldsymbol{\mu}_y), \tag{34}$$

where $\boldsymbol{\mu}_y$ and $\boldsymbol{\Sigma}_y$ represent the mean and variance of the Gaussian distribution associated with the final-layer hidden state.

# F Supplementary Experimental Results

## F.1 Hyperparameter Ablation: Smoothing Order k

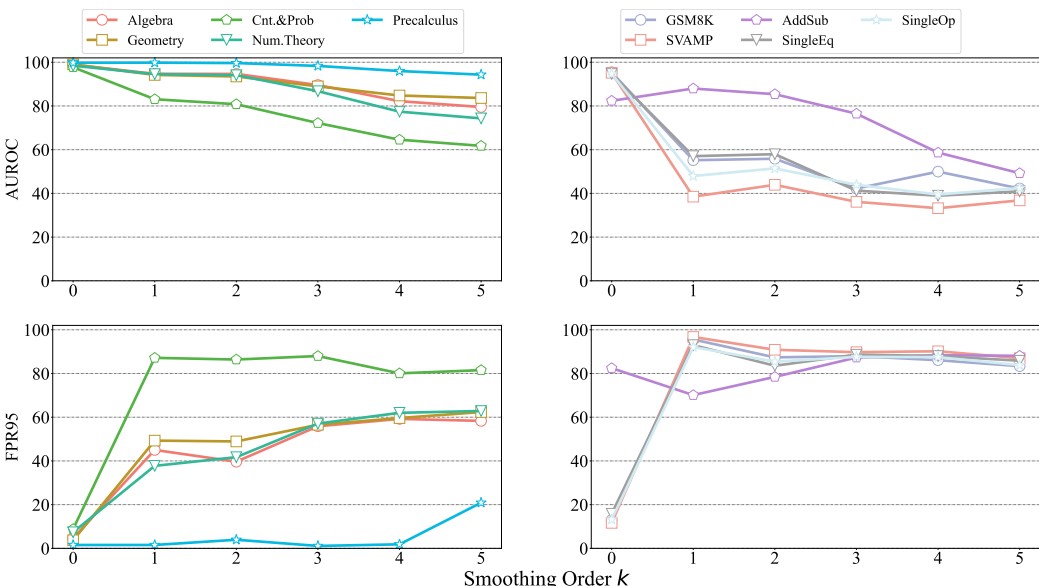

Figure 4: **Smoothing order $k$ analysis**: $k$ ranges from $0 - 5$ ($k = 0$ corresponds to the original TV Score). The upper part is for the OOD detection scenario and the lower part is for the OOD quality estimation scenario; the left part is for the far-shift OOD datasets and the right part is for the near-shift OOD datasets.

In the main experiments, we found that differential smoothing is not as effective as the basic TV score, with excellent results occurring on only a few datasets. Figure 4 visualizes the results for the

OOD detection scenario with no smoothing ($k = 0$) and smoothing order $k$ 1-5. We find that there is a significant effect of differential smoothing in two cases: (1) very good performance without smoothing, *e.g.*, the precalculus dataset, where the AUROC results are almost close to 100%; and (2) significantly poor performance without smoothing, *e.g.*, the AddSub dataset, where the FPR95 results on all other near-shift OOD datasets are 20 below, the results on this dataset are more than 80, when differential smoothing helps to eliminate some of the noise features and helps better detection.

In addition, for the case of $k > 0$, the peak performance basically occurs in the case of $k = 1$ or 2, which indicates that when $k$ is too large, the phenomenon of over-smoothing tends to occur. Too much useful feature information is lost, leading to a decrease in detection accuracy.

## F.2 OOD Detection

Results of each dataset are shown in Table 11 (Llama2-7B) and Table 12 (GPT2-XL).

Table 11: AUROC and FPR95 results in **Llama2-7B** of the **Offline Detection scenario** ($p$-value > 0.05 are grayed out). Underline and **bold** denote SOTA among all baselines and all methods.

| | | Far-shift OOD Setting | | | | |
|---|---|---|---|---|---|---|
| Dataset | Algebra | Geometry | Counting and Probability | Number Theory | Precalculus | Average |
| Method | | | AUROC ↑ / FPR95 ↓ | | | |
| Max Softmax Prob. [12] | 79.97±1.60 / 80.97±4.90 | 82.60±1.25 / 82.57±4.62 | 63.87±1.64 / 96.19±1.57 | 76.20±1.46 / 85.20±2.69 | 90.68±0.94 / 62.27±4.03 | 78.66±1.38 / 81.44±3.56 |
| Monte-Carlo Dropout [8] | 72.12±1.68 / 85.43±6.60 | 74.23±1.74 / 81.17±6.28 | 55.64±2.87 / 97.15±1.34 | 67.53±2.08 / 89.09±5.41 | 73.61±2.66 / 82.36±4.78 | 68.63±2.21 / 87.04±4.88 |
| Perplexity [3] | 84.17±1.43 / 52.75±5.13 | 88.67±1.36 / 53.32±5.38 | 77.42±1.97 / 71.28±3.90 | 83.88±2.13 / 68.90±4.22 | 94.05±0.43 / 19.03±3.17 | 85.64±1.46 / 53.06±4.36 |
| Input Embedding [43] | 81.51±1.09 / 62.84±4.56 | 75.41±0.97 / 69.66±2.07 | 62.53±1.30 / 88.35±1.72 | 84.42±0.85 / 54.80±7.25 | 75.59±0.92 / 63.71±2.87 | 75.89±1.03 / 67.87±3.69 |
| Output Embedding [43] | 76.95±1.54 / 74.92±3.02 | 78.72±1.31 / 80.97±2.23 | 61.43±1.40 / 88.45±1.58 | 70.23±1.36 / 80.18±1.71 | 86.97±1.32 / 51.51±2.27 | 74.86±1.39 / 75.21±2.16 |
| TV score (Ours) | **98.87±0.16 / 4.67±1.41** | **99.03±0.09 / 3.70±0.42** | **97.70±0.15 / 8.83±1.33** | **98.43±0.13 / 7.37±1.46** | 99.78±0.02 / 1.47±0.20 | **98.76±0.11 / 5.21±0.98** |
| w/ DiSmo (Ours) | 94.71±0.93 / 39.65±7.28 | 94.08±0.80 / 50.52±6.14 | 83.08±1.28 / 80.07±1.57 | 94.57±0.75 / 37.74±7.58 | **99.82±0.02 / 1.11±0.19** | 93.25±0.76 / 41.82±4.69 |
| Δ (**bold** - underline) | +14.70 / -48.08 | +10.36 / -49.62 | +20.28 / -62.45 | +14.01 / -47.43 | +5.77 / -17.92 | +13.12 / -47.85 |
| | | Near-shift OOD Setting | | | | |
| Dataset | GSM8K | SVAMP | AddSub | SingleEq | SingleOp | Average |
| Method | | | AUROC ↑ / FPR95 ↓ | | | |
| Max Softmax Prob. [12] | 53.08±1.67 / 94.07±1.86 | 56.56±1.53 / 90.06±2.39 | 63.31±1.88 / 87.36±2.42 | 66.68±1.32 / 86.15±2.61 | 61.07±1.30 / 86.91±2.76 | 60.14±1.54 / 88.91±2.41 |
| Monte-Carlo Dropout [8] | 48.87±2.43 / 96.78±1.21 | 44.90±2.76 / 92.33±2.09 | 57.34±1.57 / 89.15±2.34 | 54.09±2.42 / 90.07±1.96 | 56.46±1.85 / 91.21±1.83 | 52.33±2.21 / 91.92±1.89 |
| Perplexity [3] | 52.24±2.57 / 95.56±1.21 | 55.12±1.80 / 89.24±2.09 | 62.88±1.76 / 80.96±2.34 | 67.14±1.34 / 81.38±1.96 | 59.39±1.98 / 83.30±1.83 | 59.35±1.88 / 86.09±1.89 |
| Input Embedding [43] | 45.68±1.50 / 95.05±1.17 | 60.92±1.34 / 86.97±4.13 | 66.28±0.92 / 76.09±2.93 | 61.18±1.22 / 87.36±1.89 | 67.60±1.20 / 77.78±2.53 | 60.33±1.37 / 84.65±2.53 |
| Output Embedding [43] | 35.39±1.24 / 91.22±1.27 | 36.77±1.14 / 90.41±1.61 | 63.08±0.92 / 77.12±3.11 | 43.70±1.02 / 86.69±0.91 | 43.55±0.99 / 86.87±1.04 | 44.50±1.06 / 86.46±1.59 |
| TV score (Ours) | **94.88±0.25 / 14.22±1.64** | **94.51±0.20 / 12.89±1.11** | 85.84±1.06 / 82.63±2.22 | **93.97±0.24 / 17.39±1.09** | **94.00±0.20 / 14.61±0.85** | **92.64±0.39 / 28.39±1.38** |
| w/ DiSmo (Ours) | 55.21±1.91 / 95.71±0.88 | 38.24±1.53 / 87.06±0.94 | **87.06±0.94 / 71.02±4.33** | 56.66±1.34 / 93.76±1.28 | 47.46±1.32 / 92.48±1.13 | 56.99±1.41 / 88.01±1.71 |
| Δ (**bold** - underline) | +41.80 / -77.00 | +33.59 / -74.08 | +20.78 / -5.07 | +26.83 / -63.99 | +26.40 / -63.17 | +32.31 / -56.26 |

Table 12: AUROC and FPR95 results in **GPT2-XL** of the **Offline Detection scenario** ($p$-value > 0.05 are grayed out). Underline and **bold** denote SOTA among all baselines and all methods.

| | | Far-shift OOD Setting | | | | |
|---|---|---|---|---|---|---|
| Dataset | Algebra | Geometry | Counting and Probability | Number Theory | Precalculus | Average |
| Method | | | AUROC ↑ / FPR95 ↓ | | | |
| Max Softmax Prob. [12] | 72.13±1.42 / 78.35±2.35 | 64.09±1.78 / 87.42±1.24 | 68.45±1.56 / 82.35±1.57 | 69.37±1.43 / 82.09±1.90 | 78.67±0.89 / 61.23±3.02 | 70.54±1.42 / 78.29±2.02 |
| Monte-Carlo Dropout [8] | 62.41±2.10 / 88.02±1.79 | 66.63±1.82 / 82.49±1.12 | 59.32±2.03 / 92.34±1.54 | 63.88±1.92 / 86.71±1.95 | 70.19±1.47 / 73.90±1.87 | 66.18±1.87 / 84.69±1.65 |
| Perplexity [3] | 84.24±1.01 / 63.78±2.38 | 82.17±0.89 / 67.90±2.45 | 79.12±1.20 / 72.03±1.23 | 72.40±1.37 / 71.65±1.45 | 86.19±0.75 / 47.27±2.98 | 80.82±1.04 / 64.53±2.10 |
| Input Embedding [43] | 86.23±0.78 / 46.12±1.45 | 83.20±0.94 / 53.29±2.06 | 79.58±1.43 / 60.45±2.95 | 89.44±0.64 / 52.49±1.87 | 92.86±0.39 / 34.32±2.16 | 86.26±0.84 / 49.33±2.10 |
| Output Embedding [43] | 78.13±1.14 / 64.46±2.78 | 77.98±1.17 / 68.82±3.26 | 71.07±1.67 / 84.63±4.24 | 80.25±0.97 / 56.23±4.05 | 82.34±0.86 / 54.08±2.79 | 77.95±1.16 / 65.64±3.42 |
| TV score (Ours) | **98.35±0.07 / 6.24±0.24** | **97.27±0.11 / 9.23±0.35** | 85.52±0.12 / 52.51±1.78 | 92.96±0.06 / 29.86±1.26 | 93.24±0.04 / 22.68±1.14 | 93.47±0.08 / 24.10±0.95 |
| w/ DiSmo (Ours) | 93.68±0.19 / 23.24±1.45 | 94.39±0.15 / 12.17±0.79 | **95.83±0.13 / 9.86±0.46** | **99.17±0.04 / 2.42±0.22** | **99.62±0.02 / 1.75±0.13** | **96.54±0.10 / 9.89±0.61** |
| Δ (**bold** - underline) | +12.12 / -39.88 | +14.07 / -44.06 | +16.25 / -50.59 | +9.27 / -50.07 | +6.24 / -32.57 | +10.28 / -39.44 |
| | | Near-shift OOD Setting | | | | |
| Dataset | GSM8K | SVAMP | AddSub | SingleEq | SingleOp | Average |
| Method | | | AUROC ↑ / FPR95 ↓ | | | |
| Max Softmax Prob. [12] | 54.06±1.58 / 92.45±2.37 | 65.50±1.43 / 73.62±4.18 | 70.60±0.92 / 77.56±2.76 | 79.96±0.87 / 57.38±1.95 | 65.47±1.18 / 80.37±2.02 | 67.12±1.20 / 76.27±2.66 |
| Monte-Carlo Dropout [8] | 63.44±2.08 / 79.85±3.35 | 62.13±1.65 / 77.28±3.21 | 67.97±1.62 / 71.42±2.88 | 71.29±1.43 / 69.22±2.41 | 52.89±1.80 / 92.78±0.67 | 63.54±1.72 / 78.08±2.50 |
| Perplexity [3] | 72.89±1.56 / 74.63±1.52 | 70.79±1.38 / 79.65±1.21 | 67.50±1.04 / 87.76±0.92 | 87.14±0.57 / 43.09±1.23 | 70.40±1.06 / 76.80±1.45 | 73.74±1.12 / 72.39±1.27 |
| Input Embedding [43] | 87.58±1.14 / 48.45±2.06 | 84.81±1.20 / 62.75±5.68 | 80.34±0.64 / 46.62±2.17 | 81.44±0.49 / 49.05±3.24 | 81.91±0.93 / 57.65±2.67 | 83.22±0.88 / 52.90±3.16 |
| Output Embedding [43] | 82.05±1.36 / 60.44±3.96 | 81.42±2.02 / 68.43±4.38 | 72.98±0.65 / 66.72±1.77 | 80.00±1.02 / 70.23±1.87 | 79.94±0.94 / 57.68±1.60 | 79.28±1.24 / 64.70±2.72 |
| TV score (Ours) | **98.26±0.06 / 1.78±0.04** | **98.99±0.02 / 1.13±0.03** | 80.56±0.82 / 52.23±1.29 | 97.35±0.23 / 13.91±0.45 | 99.12±0.02 / 0.07±0.01 | **94.86±0.23 / 13.82±0.36** |
| w/ DiSmo (Ours) | 97.62±0.21 / 3.58±0.16 | 92.19±0.44 / 13.07±1.68 | **83.35±0.57 / 41.70±1.29** | **97.83±0.03 / 9.94±0.32** | **99.94±0.02 / 0.03±0.01** | 94.19±0.25 / 13.66±0.69 |
| Δ (**bold** - underline) | +10.68 / -46.67 | +14.18 / -61.62 | +3.01 / -4.92 | +10.69 / -33.15 | +18.03 / -57.62 | +11.64 / -39.24 |

### F.3 Beyond Detection: OOD Quality Estimation

Results of each dataset are shown in Table 13 (Llama2-7B) and Table 14 (GPT2-XL).

Table 13: **OOD Quality Estimation (Llama2-7B)**: Kendall's $\tau$ and Spearman correlation between various OOD scores and benchmark quality metric binary matching. Each column shows the correlation when ID and OOD samples are merged. Underline denotes the SOTA among all baselines, and **bold** denotes the SOTA among our methods.

| Dataset / Method | ID + Far-shift OOD | | | | | | ID + Near-shift OOD | | | | | |
|---|---|---|---|---|---|---|---|---|---|---|---|---|
| | Algebra | Geometry | Cnt.&Prob. | Num.Theory | Precalculus | Average | GSM8K | SVAMP | AddSub | SingleEq | SingleOp | Average |
| **Kendall Rank Correlation Coefficient** | | | | | | | | | | | | |
| Max Softmax Prob. [12] | $0.035_{\pm0.025}$ | $0.024_{\pm0.020}$ | $0.084_{\pm0.022}$ | $-0.043_{\pm0.021}$ | $0.021_{\pm0.014}$ | $0.024_{\pm0.020}$ | $0.064_{\pm0.022}$ | $0.067_{\pm0.015}$ | $0.076_{\pm0.016}$ | $0.057_{\pm0.019}$ | $-0.075_{\pm0.017}$ | $0.038_{\pm0.018}$ |
| Perplexity [3] | $-0.027_{\pm0.020}$ | $0.071_{\pm0.013}$ | $\underline{0.112}_{\pm0.015}$ | $0.035_{\pm0.016}$ | $0.036_{\pm0.011}$ | $0.050_{\pm0.015}$ | $\underline{0.073}_{\pm0.019}$ | $0.027_{\pm0.017}$ | $\underline{0.082}_{\pm0.015}$ | $\underline{0.091}_{\pm0.015}$ | $\underline{0.079}_{\pm0.017}$ | $\underline{0.074}_{\pm0.017}$ |
| Input Embedding [43] | $\underline{0.074}_{\pm0.020}$ | $\underline{0.119}_{\pm0.015}$ | $0.054_{\pm0.018}$ | $0.111_{\pm0.016}$ | $0.034_{\pm0.013}$ | $\underline{0.078}_{\pm0.016}$ | $0.042_{\pm0.020}$ | $-0.025_{\pm0.016}$ | $0.052_{\pm0.014}$ | $0.055_{\pm0.020}$ | $0.058_{\pm0.018}$ | $0.036_{\pm0.018}$ |
| Output Embedding [43] | $0.050_{\pm0.023}$ | $0.078_{\pm0.019}$ | $0.064_{\pm0.020}$ | $\underline{0.065}_{\pm0.020}$ | $0.034_{\pm0.009}$ | $0.058_{\pm0.018}$ | $0.042_{\pm0.018}$ | $-0.001_{\pm0.013}$ | $0.056_{\pm0.015}$ | $0.032_{\pm0.015}$ | $0.061_{\pm0.015}$ | $0.038_{\pm0.015}$ |
| TV score (Ours) | $0.182_{\pm0.016}$ | $\mathbf{0.116}_{\pm0.011}$ | $\mathbf{0.191}_{\pm0.011}$ | $0.174_{\pm0.014}$ | $0.142_{\pm0.009}$ | $\mathbf{0.161}_{\pm0.012}$ | $\mathbf{0.146}_{\pm0.010}$ | $\mathbf{0.209}_{\pm0.019}$ | $0.195_{\pm0.018}$ | $\mathbf{0.145}_{\pm0.017}$ | $\mathbf{0.101}_{\pm0.020}$ | $\mathbf{0.159}_{\pm0.017}$ |
| w/ DiSmo (Ours) | $\mathbf{0.095}_{\pm0.020}$ | $0.059_{\pm0.014}$ | $0.123_{\pm0.020}$ | $0.131_{\pm0.016}$ | $\mathbf{0.148}_{\pm0.009}$ | $0.111_{\pm0.016}$ | $0.178_{\pm0.025}$ | $0.113_{\pm0.015}$ | $\mathbf{0.121}_{\pm0.016}$ | $0.079_{\pm0.017}$ | $0.076_{\pm0.019}$ | $0.113_{\pm0.018}$ |
| Δ (**bold** - underline) | +0.021 | -0.003 | +0.079 | +0.109 | +0.112 | +0.083 | +0.073 | +0.142 | +0.039 | +0.055 | +0.022 | +0.085 |
| **Spearman Rank Correlation Coefficient** | | | | | | | | | | | | |
| Max Softmax Prob. [12] | $0.027_{\pm0.021}$ | $0.057_{\pm0.019}$ | $0.039_{\pm0.021}$ | $0.067_{\pm0.021}$ | $0.004_{\pm0.017}$ | $0.038_{\pm0.020}$ | $0.071_{\pm0.021}$ | $-0.086_{\pm0.018}$ | $0.063_{\pm0.019}$ | $0.081_{\pm0.015}$ | $0.002_{\pm0.019}$ | $0.026_{\pm0.016}$ |
| Perplexity [3] | $0.039_{\pm0.014}$ | $0.018_{\pm0.018}$ | $\underline{0.073}_{\pm0.017}$ | $0.051_{\pm0.015}$ | $0.045_{\pm0.015}$ | $0.045_{\pm0.016}$ | $\underline{0.086}_{\pm0.019}$ | $0.090_{\pm0.019}$ | $0.016_{\pm0.019}$ | $0.039_{\pm0.015}$ | $0.017_{\pm0.016}$ | $0.050_{\pm0.018}$ |
| Input Embedding [43] | $\underline{0.150}_{\pm0.021}$ | $\underline{0.136}_{\pm0.018}$ | $0.061_{\pm0.015}$ | $0.038_{\pm0.016}$ | $\underline{0.127}_{\pm0.016}$ | $\underline{0.102}_{\pm0.017}$ | $0.042_{\pm0.022}$ | $\underline{0.128}_{\pm0.016}$ | $\underline{0.154}_{\pm0.016}$ | $\underline{0.110}_{\pm0.017}$ | $\underline{0.140}_{\pm0.015}$ | $\underline{0.115}_{\pm0.017}$ |
| Output Embedding [43] | $0.002_{\pm0.018}$ | $0.001_{\pm0.020}$ | $0.015_{\pm0.015}$ | $0.033_{\pm0.016}$ | $0.012_{\pm0.018}$ | $0.025_{\pm0.017}$ | $-0.040_{\pm0.017}$ | $-0.019_{\pm0.016}$ | $0.045_{\pm0.019}$ | $0.101_{\pm0.018}$ | $-0.027_{\pm0.017}$ | $0.012_{\pm0.017}$ |
| TV score (Ours) | $0.122_{\pm0.012}$ | $\mathbf{0.169}_{\pm0.015}$ | $0.102_{\pm0.016}$ | $0.126_{\pm0.016}$ | $\mathbf{0.216}_{\pm0.014}$ | $0.147_{\pm0.015}$ | $\mathbf{0.124}_{\pm0.016}$ | $\mathbf{0.188}_{\pm0.017}$ | $0.127_{\pm0.017}$ | $\mathbf{0.165}_{\pm0.017}$ | $\mathbf{0.188}_{\pm0.019}$ | $\mathbf{0.158}_{\pm0.017}$ |
| w/ DiSmo (Ours) | $\mathbf{0.139}_{\pm0.014}$ | $0.148_{\pm0.013}$ | $\mathbf{0.176}_{\pm0.017}$ | $\mathbf{0.178}_{\pm0.017}$ | $0.121_{\pm0.016}$ | $\mathbf{0.152}_{\pm0.015}$ | $0.071_{\pm0.018}$ | $0.150_{\pm0.020}$ | $\mathbf{0.167}_{\pm0.014}$ | $0.131_{\pm0.017}$ | $0.153_{\pm0.017}$ | $0.134_{\pm0.017}$ |
| Δ (**bold** - underline) | -0.011 | +0.033 | +0.103 | +0.127 | +0.089 | +0.050 | +0.038 | +0.060 | +0.013 | +0.055 | +0.048 | +0.043 |

Table 14: **OOD Quality Estimation (GPT2-XL)**: Kendall's $\tau$ and Spearman correlation between various OOD scores and benchmark quality metric binary matching. Each column shows the correlation when ID and OOD samples are merged. Underline denotes the SOTA among all baselines, and **bold** denotes the SOTA among our methods.

| Dataset / Method | ID + Far-shift OOD | | | | | | ID + Near-shift OOD | | | | | |
|---|---|---|---|---|---|---|---|---|---|---|---|---|
| | Algebra | Geometry | Cnt.&Prob. | Num.Theory | Precalculus | Average | GSM8K | SVAMP | AddSub | SingleEq | SingleOp | Average |
| **Kendall Rank Correlation Coefficient** | | | | | | | | | | | | |
| Max Softmax Prob. [12] | $0.078_{\pm0.018}$ | $0.032_{\pm0.015}$ | $\underline{0.093}_{\pm0.014}$ | $0.059_{\pm0.017}$ | $\underline{0.068}_{\pm0.012}$ | $\underline{0.066}_{\pm0.015}$ | $0.032_{\pm0.024}$ | $\underline{0.061}_{\pm0.020}$ | $\underline{0.088}_{\pm0.018}$ | $0.023_{\pm0.017}$ | $\underline{0.081}_{\pm0.019}$ | $\underline{0.057}_{\pm0.018}$ |
| Perplexity [3] | $0.023_{\pm0.014}$ | $\underline{0.086}_{\pm0.015}$ | $0.043_{\pm0.016}$ | $-0.009_{\pm0.015}$ | $0.036_{\pm0.012}$ | $0.036_{\pm0.014}$ | $0.068_{\pm0.022}$ | $0.008_{\pm0.022}$ | $0.035_{\pm0.018}$ | $0.001_{\pm0.015}$ | $0.064_{\pm0.015}$ | $0.035_{\pm0.018}$ |
| Input Embedding [43] | $0.084_{\pm0.012}$ | $0.052_{\pm0.011}$ | $0.089_{\pm0.013}$ | $\underline{0.062}_{\pm0.011}$ | $0.007_{\pm0.010}$ | $0.059_{\pm0.012}$ | $0.041_{\pm0.019}$ | $0.010_{\pm0.020}$ | $-0.008_{\pm0.016}$ | $0.019_{\pm0.015}$ | $0.000_{\pm0.018}$ | $0.012_{\pm0.018}$ |
| Output Embedding [43] | $\underline{0.089}_{\pm0.012}$ | $0.023_{\pm0.012}$ | $0.058_{\pm0.013}$ | $0.014_{\pm0.020}$ | $0.067_{\pm0.011}$ | $0.050_{\pm0.012}$ | $\underline{0.068}_{\pm0.018}$ | $0.036_{\pm0.020}$ | $0.033_{\pm0.018}$ | $0.022_{\pm0.014}$ | $0.021_{\pm0.017}$ | $0.036_{\pm0.017}$ |
| TV score (Ours) | $0.127_{\pm0.010}$ | $\mathbf{0.142}_{\pm0.010}$ | $\mathbf{0.158}_{\pm0.009}$ | $0.098_{\pm0.011}$ | $\mathbf{0.167}_{\pm0.009}$ | $0.138_{\pm0.010}$ | $0.087_{\pm0.013}$ | $\mathbf{0.166}_{\pm0.019}$ | $\mathbf{0.110}_{\pm0.015}$ | $\mathbf{0.152}_{\pm0.013}$ | $0.141_{\pm0.016}$ | $\mathbf{0.131}_{\pm0.015}$ |
| w/ DiSmo (Ours) | $\mathbf{0.134}_{\pm0.010}$ | $0.112_{\pm0.009}$ | $0.132_{\pm0.010}$ | $\mathbf{0.161}_{\pm0.009}$ | $0.158_{\pm0.009}$ | $\mathbf{0.139}_{\pm0.009}$ | $\mathbf{0.102}_{\pm0.011}$ | $0.099_{\pm0.021}$ | $0.094_{\pm0.015}$ | $0.126_{\pm0.013}$ | $\mathbf{0.193}_{\pm0.012}$ | $0.123_{\pm0.014}$ |
| Δ (**bold** - underline) | +0.045 | +0.056 | +0.065 | +0.099 | +0.099 | +0.080 | +0.034 | +0.105 | +0.022 | +0.129 | +0.112 | +0.074 |
| **Spearman Rank Correlation Coefficient** | | | | | | | | | | | | |
| Max Softmax Prob. [12] | $0.001_{\pm0.015}$ | $0.035_{\pm0.017}$ | $0.054_{\pm0.016}$ | $0.076_{\pm0.016}$ | $\underline{0.056}_{\pm0.018}$ | $0.044_{\pm0.016}$ | $0.032_{\pm0.023}$ | $0.066_{\pm0.024}$ | $0.031_{\pm0.021}$ | $0.067_{\pm0.021}$ | $0.087_{\pm0.020}$ | $\underline{0.057}_{\pm0.022}$ |
| Perplexity [3] | $0.056_{\pm0.018}$ | $0.076_{\pm0.016}$ | $0.024_{\pm0.016}$ | $0.035_{\pm0.017}$ | $-0.002_{\pm0.016}$ | $0.038_{\pm0.017}$ | $0.023_{\pm0.021}$ | $\underline{0.094}_{\pm0.022}$ | $0.065_{\pm0.016}$ | $\underline{0.088}_{\pm0.019}$ | $0.019_{\pm0.019}$ | $0.058_{\pm0.019}$ |
| Input Embedding [43] | $\underline{0.141}_{\pm0.016}$ | $\underline{0.135}_{\pm0.016}$ | $0.087_{\pm0.014}$ | $\underline{0.097}_{\pm0.017}$ | $0.029_{\pm0.016}$ | $\underline{0.098}_{\pm0.016}$ | $0.043_{\pm0.018}$ | $0.084_{\pm0.020}$ | $0.098_{\pm0.016}$ | $0.024_{\pm0.016}$ | $0.090_{\pm0.017}$ | $0.068_{\pm0.016}$ |
| Output Embedding [43] | $0.012_{\pm0.018}$ | $-0.098_{\pm0.016}$ | $0.076_{\pm0.016}$ | $0.054_{\pm0.017}$ | $0.035_{\pm0.017}$ | $0.016_{\pm0.017}$ | $0.010_{\pm0.019}$ | $0.059_{\pm0.025}$ | $-0.077_{\pm0.019}$ | $0.084_{\pm0.019}$ | $0.067_{\pm0.023}$ | $0.029_{\pm0.021}$ |
| TV score (Ours) | $0.165_{\pm0.013}$ | $0.110_{\pm0.012}$ | $\mathbf{0.141}_{\pm0.012}$ | $0.086_{\pm0.015}$ | $\mathbf{0.115}_{\pm0.014}$ | $0.123_{\pm0.013}$ | $0.112_{\pm0.012}$ | $\mathbf{0.164}_{\pm0.012}$ | $0.125_{\pm0.018}$ | $\mathbf{0.168}_{\pm0.018}$ | $\mathbf{0.161}_{\pm0.014}$ | $0.146_{\pm0.015}$ |
| w/ DiSmo (Ours) | $\mathbf{0.173}_{\pm0.011}$ | $\mathbf{0.208}_{\pm0.010}$ | $0.072_{\pm0.015}$ | $\mathbf{0.145}_{\pm0.019}$ | $0.109_{\pm0.016}$ | $\mathbf{0.141}_{\pm0.014}$ | $\mathbf{0.134}_{\pm0.015}$ | $0.127_{\pm0.016}$ | $\mathbf{0.189}_{\pm0.017}$ | $0.167_{\pm0.017}$ | $0.155_{\pm0.016}$ | $\mathbf{0.154}_{\pm0.016}$ |
| Δ (**bold** - underline) | +0.032 | +0.073 | +0.054 | +0.048 | +0.059 | +0.043 | +0.091 | +0.070 | +0.091 | +0.080 | +0.071 | +0.086 |

### F.4 Beyond Mathematical Reasoning

Our method has a wider range of application scenarios beyond mathematical reasoning. To verify generalizability, we choose the multiple-choice quizzing task, which has the same "pattern collapse" property as mathematical reasoning, since the output space is limited to the "ABCD" four options.

We select the MMLU dataset [10] and choose eight domains among it: high school mathematics, high school physics, high school chemistry, high school biology, high school geography, high school government and politics, high school psychology, high school statistics. We test eight rounds, each using one of the domains as the ID dataset and the remaining seven domains as the OOD datasets.

We use Llama2-7B as the training backbone, each model is trained for 3K steps and 8 batch sizes in 4-card NVIDIA Tesla V100 (2 per card). The AUROC score matrices are shown in Figure 5(a)-(e),

presenting the results for TV score, input embedding, output embedding, and perplexity, respectively. In the figures, the Roman numeral I - VIII are represented as:

- I = high school mathematics
- II = high school physics
- III = high school chemistry
- IV = high school biology
- V = high school geography
- VI = high school government and politics
- VII = high school psychology
- VIII = high school statistics

We find that MS-Prob and PPL nearly fail on the multiple-choice task and the output embedding is not as excellent as expected, which is caused by the pattern collapse phenomenon.

Our method is comparable to the input embedding method and has very good absolute performances. For some far-shift OOD scenarios, *e.g.*, mathematics-psychology (I-VII), physics-politics (II-VI), etc., performances of the input embedding method and our method are basically the same, and there exists a reasonable range of competing phenomena, *e.g.*, our method performs more advantageously under physics-biology (II-IV), and the input embedding method is better under physics-geography (II-V). For some near-shift OOD scenarios, *e.g.*, mathematics-statistics (I-VIII), where both domains essentially belong to the category of math, our method will be more well-performed, indicating that the embedding-based method produces performance degradation in fine-grained scenarios, while our method possesses stronger robustness.

Overall, our method is scalable and has greater advantages in fine-grained detection scenarios.

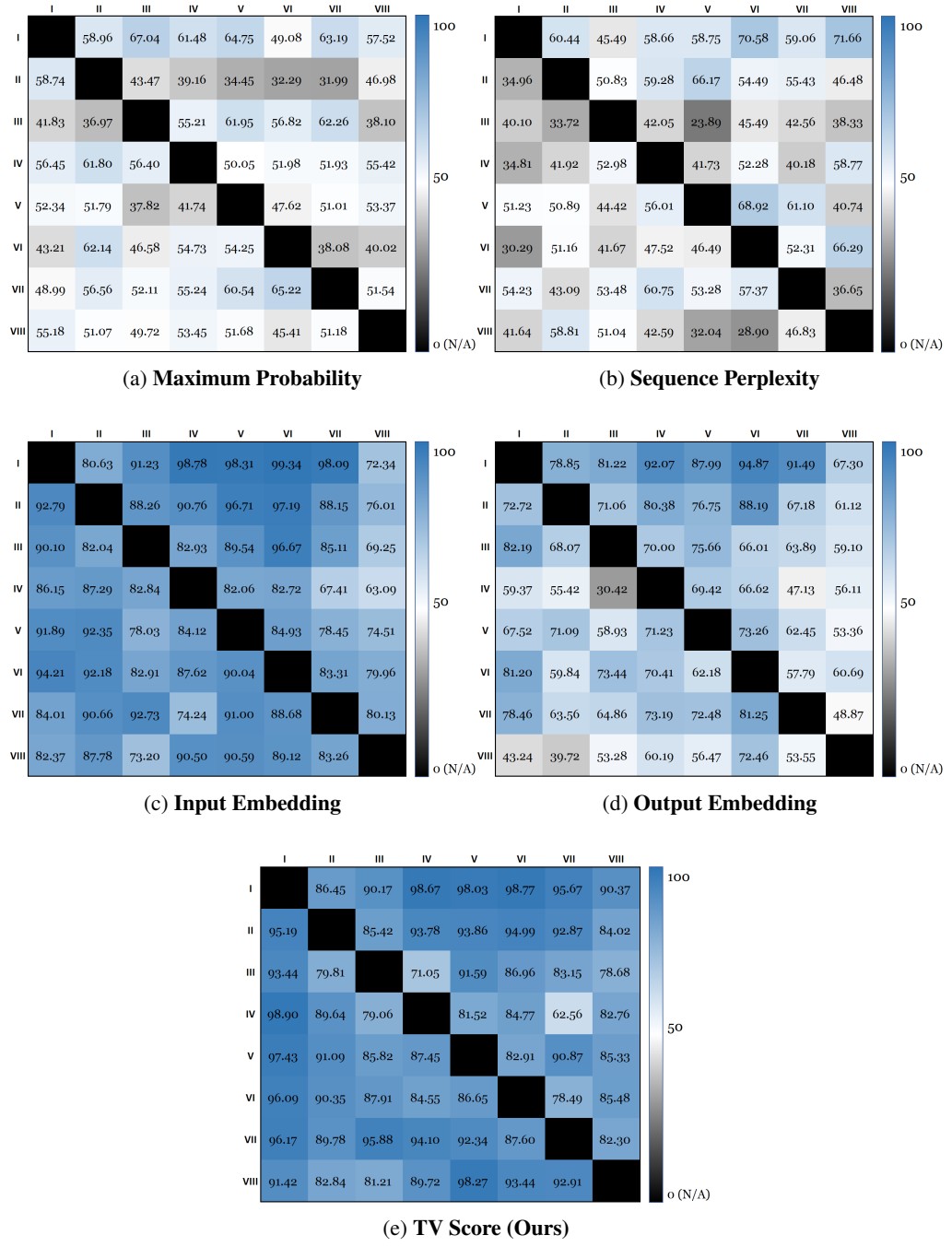

Figure 5: AUROC score matrix in MMLU dataset of different OOD scores. Rows represent ID data, and columns represent OOD data.

