# OpenReview forum: "Embedding Trajectory for Out-of-Distribution Detection in Mathematical Reasoning"
_NeurIPS.cc/2024/Conference — NeurIPS 2024 poster_

### Official Review · Reviewer_oehe · 2024-07-12

**Soundness:** 3
**Presentation:** 3
**Contribution:** 3
**Rating:** 7
**Confidence:** 2

**Summary:**

This work studied the ODD detection in mathematical reasoning, which presents a new measurement, TV score, based on the observed pattern collapse property and early stabilization of GLMs. It seems the first discussion on the OOD detection in mathematical reasoning.

**Strengths:**

This appears to be the first discussion on OOD detection in mathematical reasoning. The authors clearly explain why the trajectory works in an understandable and empirical manner. The manuscript is well-organized, and the methodology is both simple and effective.

**Weaknesses:**

1. Additional metrics commonly used for ODD should be included for evaluation.
2. The writing requires improvement.
3. Further insight into the potential impact of over-smoothing on the setting of the critical parameter $k$ is necessary.

**Questions:**

1. Is it true that the embedding dimension should be fixed across layers to compute the embedding difference between neighboring layers?
2. To better support the claim of choosing $k \leq 5$, it would be beneficial to provide a quantitative analysis of over-smoothing occurrence when $k$ is set to a larger value.
3. For testing the ODD performance, could you report the evaluation results in terms of AUPR and F1?
4. Which datasets were used for the analysis in Figure 3?
5. There are some grammatical mistakes that need fixing. Here are a few examples:
   (a) "we need ..., then computer ..." in Line 150.
   (b) The sentence in Lines 161-162.
   (c) "Outliers ..., then lead ..." in Lines 164-165.

**Limitations:**

The authors adequately claimed the limitations of this work.

---

> ### Author Rebuttal · Authors · 2024-08-04
>
> **Thanks for your constructive comments! We will respond to the weaknesses and questions you raised in the following areas:**
>
> ---
>
> > More Metrics (W1 & Q3)
>
> Thank you for suggesting richer evaluation metrics about AUPR and F1. The results are below:
>
> | Method | Llama2-7B | | | |  GPT2-XL | | | |
> | :--- | :---: | :---: | :---: | :---: | :---: | :---: | :---: | :---: |
> | | Far-shift OOD | | Near-shift OOD | | Far-shift OOD | | Near-shift OOD |
> | | AUPR | F1 | AUPR | F1 | AUPR | F1 | AUPR | F1 |
> | MS-Prob | 74.22±1.20 | 68.42±1.39 | 53.30±1.65 | 66.86±1.06 | 81.35±0.98 | 71.78±0.98 | 66.26±0.99 | 65.45±0.85 |
> | MC-Drop | 57.86±0.34 | 54.87±1.92 | 40.87±0.66 | 59.78±1.29 | 65.13±1.09 | 62.34±1.38 | 63.24±0.87 | 61.12±1.33 |
> | PPL | 78.89±0.40 | 72.25±0.87 | 62.73±0.98 | 75.58±1.11 | 71.32±1.21 | 75.60±0.74 | 62.64±0.59 | 60.95±1.16 |
> | I-Emb | 79.63±1.17 | 69.08±1.11 | 54.72±1.89 | 61.92±0.99 | 86.69±0.37 | 88.42±0.29 | 78.82±0.62 | 69.29±1.09 |
> | O-Emb| 60.58±1.25 | 56.06±1.65 | 39.84±0.81 | 60.05±0.94 | 78.24±0.86 | 85.34±0.41 | 75.39±0.75 | 68.83±1.06 |
> | TV score (ours) | **98.81±0.16** | **93.17±0.49** | **90.73±0.42** | **80.43±0.54** | 98.04±0.08 | 98.86±0.09 | **89.91±0.65** | **78.15±0.76** |
> | w/ DiSmo (ours) | 96.19±0.06 | 85.79±0.78 | 80.57±0.72 | 78.98±0.65 | **98.28±0.06** | **99.63±0.07** | 84.04±0.70 | 76.03±0.94 |
>
> **On both metrics, our approach still maintains a significant lead.**
>
> ---
>
> > Discussion about k (W3 & Q2)
>
> Thanks for suggesting a more detailed discussion of $k$. In Section 5 and Appendix G.1, we have conducted ablation studies for cases of $k \leq 5$. To support our claim of choosing $ k \leq 5$ and the evidence about the over-smoothing phenomenon when $k$ is too large, we report more AUROC results in Llama2-7B and GPT2-XL as we continue to increase $k$ values:
>
> | $k$ value | Llama2-7B | | GPT2-XL | |
> | --- | --- | --- | --- | --- |
> | | Far-shift OOD | Near-shift OOD | Far-shift OOD | Near-shift OOD
> | 0 | 98.76 | 92.64 | 93.47 | 94.86 |
> | 1 | 94.71 | 87.98 | 95.55 | 94.08 |
> | 2 | 94.66 | 85.39 | 96.54 | 94.19 |
> | 3 | 89.57 | 76.47 | 95.32 | 93.44 |
> | 4 | 82.20 | 58.66 | 95.17 | 92.09 |
> | 5 | 79.52 | 49.25 | 94.26 | 92.18 |
> | 6 | 57.65 | 47.89 | 93.01 | 82.65 |
> | 7 | 55.27 | 47.16 | 90.28 | 76.50 |
> | 8 | 58.63 | 48.34 | 82.11 | 76.18 |
> | 9 | 54.10 | 51.28 | 73.89 | 68.22 |
> | 10 | 52.93 | 49.72 | 59.66 | 57.91 |
>
> When $k>5$, the AUROC value on Llama2-7B shows a sharp drop and stabilizes around 50. The AUROC value on GPT2-XL remains high, but also shows a sharp decrease when $k$ increases close to 10. This is because GPT2-XL has 1.5 times more layers than Llama2-7B, the trajectory information is richer and the noise will be relatively more.
>
> Overall, as $k$ continues to increase, the useful trajectory volatility information is gradually blurred even though more noise is erased, causing over-smoothing. Therefore, $k \leq 5$ is a better trade-off range, this is why we choose $k \leq 5$.
>
> ---
>
> > Experimental Setup about Figure 3 (Q4)
>
> The ID data curve is the average of all samples in the MultiArith dataset, and the OOD data curve is the average of all samples in the five domains of the MATH dataset (Algebra, Geometry, Counting and Probability, Number Theory, and Precalculus). We will add the setup in the updated version.
>
> ---
>
> > Method Detail: Fixed Embedding Dimension (Q1)
>
> The value of the embedding dimension is arbitrary and does not need to be fixed, but the embedding dimensions of neighboring layers should be equal, otherwise they cannot be subtracted.
>
> For language models, the output dimension of the hidden layer is the embedding dimension set in the model configuration (e.g., 4096 for Llama2-7B and 1600 for GPT2-XL), so there is no problem of inconsistent dimensionality.
>
> ---
>
> > Paper Writing (W2 & Q5)
>
> Thank you for pointing out the mistakes in our writing, we will correct them in the updated version.
>
> ---
>
> **We expect the above responses to address your concerns, and look forward to your more positive comments.**

---

> > ### Comment · Reviewer_oehe · 2024-08-13
> >
> > Thanks for your efforts to address my concerns. I stay positive for this work.

---

> > > ### Author Response · Authors · 2024-08-13
> > > **Thanks for recognition**
> > >
> > > Thank you for supporting our work!

---

### Official Review · Reviewer_A7h4 · 2024-07-13

**Soundness:** 2
**Presentation:** 2
**Contribution:** 2
**Rating:** 3
**Confidence:** 3

**Summary:**

This paper presents a trajectory-based method for OOD detection in the mathematical reasoning setting. OOD detection is extensively studied in the text setting and image setting. The main motivation of this work is claimed to that mathematical reasoning poses significant challenges to embedding-based methods due to its high-density feature of output spaces, but this feature causes larger discrepancies in the embedding shift trajectory between different samples in latent spaces. The proposed method uses trajectory volatility for OOD detection in mathematical reasoning. Experiments are conducted to validate the performance of the proposed method.

**Strengths:**

This paper studies OOD detection in the mathematical reasoning setting, which is less studied compared to text and image setting.

This paper uses examples to illustrate the motivation and key idea. This improves the accessibility of this paper.

**Weaknesses:**

The motivation is not strong and convincing. Figure 1 illustrates two challenges with respect to the input space and out space respectively. It is not convincing, and more evidence or data analysis should be provided to make these two challenges solid. I do not buy that pattern collapse is generally hold for all types of mathematical reasoning tasks. The example is just a special case. The output can take any value in the real line, and the collapse probability not that high.

The idea of the proposed method is not convincing. I am not convinced that early stabilization is generally held for mathematical reasoning problems. More evidence should be provided to justify it.

The writing is also not precise. For example, in Equation (1), the domain of f is not specified. The notation \phi is not defined.

Also, the experiment is not solid. There is no experiment to justify that the performance improvement is due to addressing the challenges in the input space and output space posed in the Introduction.

**Questions:**

See weakness.

**Limitations:**

See weakness.

---

> ### Author Rebuttal · Authors · 2024-08-05
>
> **Thanks for your constructive comments! We will respond to your concerns one by one.**
>
> ---
>
> > W1: The motivation is not strong and convincing. Figure 1 ...
>
> ### 1. Input Space
>
> As for the phenomenon that embedding varies less in different domains of input space: We have discussed it in Section 5 and conducted experiments in Appendix G.2 to demonstrate this.
>
> ### 2. Output Space
>
> Please refer to **"General Rebuttal: Existence and Universality of "Pattern Collapse" phenomenon in mathematical reasoning"** for detailed responses and evidences.
>
> We must claim a key fact: **For generative language models (GLMs), they model real numbers or mathematical expressions not in a mathematical sense, but based on discrete token sequences after tokenization. Thus, the collapse occurs at the token level, not at the full mathematical expression level**. Due to the autoregressive generative nature of GLMs, the collapse phenomenon occurs during the prediction of each token.
>
> We agree with your opinion that the number of values on the real line is infinite in a mathematical sense. However, **after tokenization, they contain only 0-9 number tokens and a limited number of special symbols**, such as decimal points, slashes, root signs. This means that two largely different expressions in the mathematical sense may cover many of the same tokens.
>
> We emphasize two key conclusions from the statistic data presented in General Rebuttal:
>
> * Existence: **The average token duplication rate is up to 99% on all math tasks, and even a staggering 99.9% on some simple arithmetic tasks**; In contrast, **the token duplication rate on the text generation task is only about 60%, with about 2000 different types of token**, and still increasing as the total number of tokens increases. *These data and comparisons demonstrate that pattern collapse occurs in mathematical reasoning and not in text generation.*
>
> * Universality: **Token repetition rate exceeded 97% on all seven math tasks of different difficulties and types.** *This demonstrates the universality of "pattern collapse" in various mathematical reasoning tasks.*
>
> ---
>
> > W2: I am not convinced that early stabilization is generally held for mathematical reasoning problems. More evidence should be provided to justify it.
>
> ### 1. Setup Description of Figure 3
>
> The ID data curve is the average of all samples in the MultiArith dataset, and the OOD data curve is the average of all samples in the five domains of MATH dataset (Algebra, Geometry, Counting and Probability, Number Theory, and Precalculus). These are consistent with our settings for the ID and OOD datasets in the experimental setup (Section 4.1). We will add this setup in the updated version.
>
> ### 2. More evidences about "early stabilization is generally held for mathematical reasoning problems"
>
> Please refer to **"General Rebuttal: Universality of "Early Stabilization" phenomenon in mathematical reasoning (Expanded visualization of Figure 3 in the paper)"** for detailed responses and evidences.
>
> ---
>
> > W3: The writing is also not precise. ...the domain of f is not specified. The notation \phi is not defined.
>
> Thanks for pointing out some questions about paper writing.
>
> * The domain of $f(\boldsymbol{x}, \boldsymbol{y}, \boldsymbol{\theta})$ is any value $(\boldsymbol{x}, \boldsymbol{y}, \boldsymbol{\theta})$ sampled from the joint probability distribution $P_{\mathcal{X} \times \mathcal{Y} \times \Theta}$ defined on the $\mathcal{X} \times \mathcal{Y} \times \Theta$, where $\mathcal{X}, \mathcal{Y}, \Theta$ are the input space, the output space, and the parameter space, respectively, as defined in lines 70-75. We did not specify this because there are no special values that need to be excluded.
>
> * The notation "\phi" does not appear in our paper.
>
> We will improve our presentation in updated version.
>
> ---
>
> > W4: The experiment is not solid. There is no experiment to justify that the performance improvement is due to addressing the challenges posed in the Introduction.
>
> ### 1. Response to "There is no experiment to justify that performance improvement is due to addressing the challenges ..."
>
> * First, **the two "challenges" are two objectively existing phenomena in mathematical reasoning scenarios. However, they make existing embedding-based methods that cannot be applied to mathematical reasoning, so we call them "challenges" for existing methods, not for us**. As stated in lines 38-39, *“However, embedding-based methods encounter challenges under mathematical reasoning scenario”*. These two challenges are simply introduced to explain why existing methods cannot be applied to mathematical reasoning;
>
> * Second, because existing methods are hindered by these two phenomena, we need to find new methods to circumvent encountering them. Thus, **we aim to circumvent the two challenges faced by existing methods, not to address them**. As stated in lines 47-48, *“Therefore, we transform our perspective from the static embedding representation to the dynamic embedding shift trajectory in the latent space"*, **we are not improving on existing methods, but rather two different research ideas.**
>
> * Third, these two phenomena are challenges for existing methods, but opportunities for us: we utilize them to design trajectory-based methods (Section 2-3).
>
> ### 2. Reasons for performance improvement
>
> A complete explanation of why our method yields performance improvement is given in Section 2 (motivation part).
>
> Refer to **"General Rebuttal: Motivation line from "pattern collapse" to "early stabilization" and TV score"** for details.
>
> ### 3. Response to “The experiment is not solid”
>
> We have performed rich dataset experiments, scalable experiments, ablation analyses, and failure analyses in Sections 4-5 and Appendices E-G to demonstrate the solidity of our experiments.
>
> ---
>
> **We expect these responses to address your concerns, and look forward to your more positive comments.**

---

> > ### Comment · Reviewer_A7h4 · 2024-08-12
> >
> > Many thanks for the clarification.  After reading the response, most of my concerns still remain such as consolidate the "Pattern Collapse" phenomenon, experiment finding, evaluation, etc.  This paper presents some interesting idea, but it needs many ideas to make it solid. Given this, I like to maintain my score.

---

> > > ### Author Response · Authors · 2024-08-13
> > > **Thanks for response & Invite open discussion**
> > >
> > > Thanks for your response. Regarding *"consolidate the "Pattern Collapse" phenomenon, experiment finding, evaluation, etc."*, we have already provided a detailed response in the rebuttal stage, including clarification, statistics, visualizations, and quotations. Here we present some key evidence and conclusions again:
> > >
> > > ---
> > >
> > > ### 1. "Pattern Collapse" phenomenon (Evidence: Fact Clarification & Statistics)
> > >
> > > First, **we have clarified that the language model's modeling of mathematical expressions is based on tokenization**, so "pattern collapse" occurs at the discrete token level, not at the level of mathematical sense. Understanding "pattern collapse" in terms of the real number line is misconceived, and it goes against the way language models are modeled.
> > >
> > > Second, **we have given statistics** on seven different types of math tasks with different domains and difficulty levels, and have compared them to two classic text generation tasks, translation and summarization.
> > > We mainly focus on the number of token types and the duplication rate, which can reflect how much the model collapses at the token level when predicting. We again present the results as follows:
> > >
> > > |Types of Tasks|token number in dataset |token type number in dataset |Token Duplication Rate|Vocab Coverage|
> > > |---|---|---|---|---|
> > > |***Mathematical Reasoning***|||||
> > > |Arithmetic (primary difficulty)|16136|14|99.9%|0.04%|
> > > |Arithmetic (middle-school difficulty)|5663|16|99.7%|0.05%|
> > > |Algebra|5234|107|98.0%|0.33%|
> > > |Geometry|2615|75|97.1%|0.23%|
> > > |Counting and probability|2524|43|98.3%|0.13%|
> > > |number theory|2395|71|97.1%|0.22%|
> > > |precalculus|3388|84|97.5%|0.26%|
> > > |*Average* | *5422* | *58* | *98.9%*| *0.18%*|
> > > |***Text Generation***||||
> > > |Translation|2500 |1065 | 57.4% | 3.32% |
> > > ||5000 | 1832| 63.3% | 5.10% |
> > > ||10000 | 2980 | 70.2% | 9.31% |
> > > || 15000 | 3494 | 76.7% | 10.61% |
> > > |*Average* | *5833* | *1959* | *66.4%*| *6.12%*|
> > > |***Summarization***||||
> > > ||2500 |1265 | 49.4% | 4.01% |
> > > ||5000 | 1970 | 60.6% | 6.16% |
> > > ||10000 | 3192 | 68.0% | 9.98% |
> > > ||15000 | 3876 | 74.1% | 12.11% |
> > > |*Average* | *5833* | *2142* | *63.2%* | *6.69%*|
> > >
> > > **The average token duplication rate is up to 99% on all math tasks, and even a staggering 99.9% on some simple arithmetic tasks; The token duplication rate exceeded 97% on all seven math tasks of different difficulties and types.
> > > This demonstrates that the "pattern collapse" occurs on generally all types of mathematical reasoning tasks.**
> > >
> > > ---
> > >
> > > ### 2. "Early Stabilization" finding (Evidence: Visualization)
> > >
> > > In our GENERAL REBUTTAL, **we have given detailed visualizations** of trajectories across ten datasets of different domains and different difficulty levels in mathematical reasoning.
> > >
> > > We have also given the average volatility statistics for layers 1-31 (full layers), 20-31, and 26-31 on each dataset corresponding to the visualizations as follows:
> > >
> > > |Dataset |1-31 layers |20-31 layers | 26-31 layers|
> > > | --- | --- | --- | --- |
> > > | *ID Dataset* | | | | |
> > > | MultiArith | 6.53 | 14.84 | 10.89 |
> > > | *Near-shift OOD Dataset* | | | | |
> > > | GSM8K | 8.60 | 20.55| 26.43 |
> > > | SVAMP | 8.02 | 18.82 | 24.68 |
> > > | AddSub | 8.72 | 20.54 | 27.24 |
> > > | SingleEq | 8.14 | 19.26 | 22.42 |
> > > | SingleOp | 7.50 | 17.17 | 21.62 |
> > > | *Far-shift OOD Dataset* | | | | |
> > > | MATH-Algebra | 8.83 | 21.35 | 31.50 |
> > > | MATH-Geometry | 10.00 | 25.27 | 34.14 |
> > > | MATH-Count_and_Prob | 10.30 | 25.77 | 33.70 |
> > > | MATH-Number_Theory | 9.40 | 23.10 | 33.86 |
> > >
> > > **In all ten datasets, the phenomenon of "early stabilization" is significantly present, and it can be sufficiently demonstrated that "early stabilization" is universal in mathematical reasoning.**
> > >
> > > ---
> > >
> > > ### 3. Experiment Justification (Evidence: Quotation)
> > >
> > > **We have explained your misunderstanding** about the purpose of our paper by quoting from the original text:
> > >
> > > * The two "challenges" are faced by existing embedding-based methods that cannot be applied to mathematical reasoning, not by us. *As stated in lines 38-39, “However, embedding-based methods encounter challenges under mathematical reasoning scenario”.*
> > >
> > > * We aim to circumvent the two challenges faced by existing methods, not to address them. We are not improving on existing methods, but rather two different research ideas. *As stated in lines 47-48, “Therefore, we transform our perspective from the static embedding representation to the dynamic embedding shift trajectory in the latent space".*
> > >
> > > ---
> > >
> > > We strongly respect your concerns about the evidence we gave, and an open discussion always helps us to recognize our limited considerations. Unfortunately, we have not received targeted details of your concerns, can you tell us what it is about our evidence that you do not recognize? Your proposed valuable issues will largely help us to improve our paper quality, and we look forward to a more open discussion with you, thanks!

---

> > > > ### Comment · Reviewer_A7h4 · 2024-08-14
> > > >
> > > > Thank you for your reply. I have no further questions.

---

### Official Review · Reviewer_KF5B · 2024-07-15

**Soundness:** 3
**Presentation:** 3
**Contribution:** 3
**Rating:** 6
**Confidence:** 4

**Summary:**

This paper studies the OOD problem in GLMs under mathematical reasoning and found that the the patter collapse phenomena in the output space. The trajectory violation score is proposed to distinguish the ID and OOD samples. A thorough evaluation shows that the proposal can outperform traditional algorithms under offline detection, online detection, and quality estimation.

**Strengths:**

(+) This paper is well structured and well-written.
(+) The authors conduct a thorough evaluation of the proposed methods

**Weaknesses:**

(-) The input data for the empirical study in Figure 3 is not introduced. Thus, whether it can reflect the scenario of mathematical reasoning is not clear. It lacks theoretical analysis.
(-) The relationship of the pattern collapse (in Figure 1) and the early stabilization (in Figure 3) is not clearly illustrated.
(-) The datasets for experiments are quite limited.

**Questions:**

1.  What is the internal relationship between the pattern collapse (in Figure 1) and the early stabilization (in Figure 3)?

**Limitations:**

The limitations are adequately addressed.

---

> ### Author Rebuttal · Authors · 2024-08-05
>
> **Thanks for your constructive comments. We will respond to your concerns one by one.**
>
> ---
>
> > W1: The input data for the empirical study in Figure 3 is not introduced. Thus, whether it can reflect the scenario of mathematical reasoning is not clear.
>
> Thanks for pointing this out, and sorry for the missing experimental setup (i.e., input data source) for Figure 3.
>
> ### 1. Setup Description of Figure 3
>
> The ID data curve is the average of all samples in the MultiArith dataset, and the OOD data curve is the average of all samples in the five domains of the MATH dataset (Algebra, Geometry, Counting and Probability, Number Theory, and Precalculus). These are consistent with our settings for the ID and OOD datasets in the experimental setup (Section 4.1). We will add this setup in the updated version.
>
> ### 2. More evidences about "it can reflect the scenario of mathematical reasoning"
>
> Please refer to **"General Rebuttal: Universality of "Early Stabilization" phenomenon in mathematical reasoning (Expanded visualization of Figure 3 in the paper)"** for detailed responses and evidences.
>
>
> ---
>
> > W2 & Q1: The relationship of the pattern collapse (in Figure 1) and the early stabilization (in Figure 3) is not clearly illustrated.
>
> Please refer to **"General Rebuttal: Motivation line from "pattern collapse" to "early stabilization" and TV score"** for detailed responses.
>
> Thanks for pointing out the vague elaboration on the motivation line. The pattern collapse (in Figure 1) and the early stabilization (in Figure 3) are not directly related, but indirectly transitioned through the theoretical intuition of Section 2.1. To summarize, the "pattern collapse" leads to more significant trajectory differences across different samples, and the source of the trajectory differences between ID and OOD samples is the "Early Stabilization" phenomenon.
>
> In detail, the function of each section is:
>
> * In Section 1, we **find the "pattern collapse" (in Figure 1)** in the output space;
> * In Section 2.1, we introduce the intuition: **the presence of "pattern collapse"** causes the convergence of the trajectory endpoints of different samples, **leading to significant trajectory differences across samples  (As stated in Lines 104 Hypothesis 1)**. We justify this intuition through theoretical modeling and proving.
> * We already have the intuition that trajectories can be a good measure, but it is still unknown what kind of difference exists between the trajectories of the ID and OOD samples. Thus, in Section 2.2, we conducted empirical experiments to observe the trajectory volatilities of different scenarios and discover the phenomenon of **"early stabilization" (in Figure 3), which is the root cause of the trajectory differences.**
>
> We will state this motivation line at the start of Section 2 in the updated version.
>
> ---
>
> > W3: The datasets for experiments are quite limited.
>
> Thanks for pointing out the limited datasets. We illustrate this in terms of both data type and data size:
>
> ### 1. Data Type
>
> As for the data type, **we have collected 10 of the most commonly used datasets** in the LLM mathematical reasoning research for our experiments, and **they cover six types of mathematical tasks and all levels of difficulties from elementary school to college**. Due to space limits, we reported the average results for each setting in the main text, and the results on each dataset have been shown in Appendix E (Tables 8-11). Overall, our dataset types have been as rich as possible.
>
> ### 2. Data Size
>
> Data size is a limitation as we mentioned in Section Limitation. However, this is caused by the particular nature of the mathematical reasoning research field, and is not a subjective constraint on us, for the following reasons:
> * Compared to traditional text generation tasks such as summarization and translation, mathematical reasoning tasks did not receive much attention prior to the era of LLMs, and thus there are few general-purpose datasets. At the same time, automated construction of mathematical datasets needs to be based on complex rules, resulting in complex mathematical tasks requiring significant and time-consuming human intervention[1,2], making diversity and size severely limited.
> * After the advent of the Chain-of-Thought technique[3], mathematical reasoning started to receive more attention from the NLP field, but some of the more complex mathematical tasks are usually built for LLM evaluation[4], such as TheoremQA (600 samples)[5], SAT-Math (220 samples)[6], MMLU-Math (974 samples)[7], making the large-scale training sets not specifically constructed.
> All these factors make the dataset size in the field of mathematical reasoning much smaller than traditional generative tasks.
>
> On the other hand, **we are the first to study OOD detection on mathematical reasoning, and the current data type is sufficient to demonstrate the generality of our method.** If new large-scale mathematical reasoning datasets become available, our method can be further applied.
>
> [1] Solving general arithmetic word problems, EMNLP 2015.
>
> [2] Measuring mathematical problem solving with the math dataset, NeurIPS 2021.
>
> [3] Chain-of-Thought Prompting Elicits Reasoning in Large Language Models, NeurIPS 2022.
>
> [4] MAmmoTH: Building Math Generalist Models through Hybrid Instruction Tuning, ICLR 2024.
>
> [5] TheoremQA: A Theorem-driven Question Answering Dataset. ACL 2023.
>
> [6] Agieval: A human-centric benchmark for evaluating foundation models. NAACL 2024.
>
> [7] Measuring massive multitask language understanding. ICLR 2021.
>
> ---
> **We expect the above responses to address your concerns, and look forward to your more positive comments.**

---

> > ### Comment · Reviewer_KF5B · 2024-08-13
> >
> > Thanks for the clarification, and my concerns have been mostly addressed. As it is a very important research problem and many illustrations and experiments have been supplemented, I'm happy to increase my score.

---

> > > ### Author Response · Authors · 2024-08-13
> > > **Thanks for recognition**
> > >
> > > Glad to see that we have addressed most of your concerns and thanks for your recognition of our work!

---

### Official Review · Reviewer_SAjz · 2024-07-18

**Soundness:** 3
**Presentation:** 3
**Contribution:** 3
**Rating:** 5
**Confidence:** 3

**Summary:**

This work discusses a novel method for out-of-distribution (OOD) detection in generative language models (GLMs), particularly in the context of mathematical reasoning tasks. The key insights are: 1) The high-density output space in mathematical reasoning leads to a "pattern collapse" that causes larger discrepancies in the embedding shift trajectory between different samples in latent spaces, 2) GLMs exhibit early stabilization for in-distribution (ID) samples in mathematical reasoning, while OOD samples do not show this behavior.

**Strengths:**

The authors propose a novel trajectory-based method called "TV score" that leverages the unique characteristics of the high-density output space in mathematical reasoning tasks to effectively detect OOD samples. This approach goes beyond the traditional OOD detection methods focused on uncertainty estimation and embedding distance measurement, which struggle in the challenging mathematical reasoning domain.

Robust OOD detection is crucial for the real-world deployment of generative language models, as these models are susceptible to performance degradation when faced with out-of-distribution inputs. The authors' work addresses a practical and important problem in the field, as mathematical reasoning tasks are increasingly incorporated into language models with high-stakes applications.

The authors' analysis of the unique characteristics of the input and output spaces in mathematical reasoning tasks provides valuable theoretical insights into the challenges posed by this domain for OOD detection.

**Weaknesses:**

The underlying mechanism of the TV score method and its relationship to the observed "pattern collapse" in the output space is not fully explained. Providing a more detailed analysis and visualization of the trajectory dynamics, as well as the intuition behind the choice of trajectory volatility as the detection metric, would enhance the interpretability of the approach.

The computational complexity of the TV score method is not discussed, which is an important consideration for real-world deployment, especially in resource-constrained environments. Investigating the computational efficiency of the method and exploring potential optimizations or approximations would be valuable for improving its practical applicability.

The paper does not address the robustness of the TV score method to adversarial attacks, which is a critical consideration for the security of generative language models. Evaluating the method's performance under different types of adversarial perturbations and developing strategies to improve its robustness would be a valuable extension of this work.

**Questions:**

What is the underlying mechanism of the TV score method, and how does it relate to the observed "pattern collapse" in the output space? How can providing a more detailed analysis and visualization of the trajectory dynamics, as well as the intuition behind the choice of trajectory volatility as the detection metric, enhance the interpretability of the approach?

How does the computational complexity of the TV score method impact its real-world deployment, especially in resource-constrained environments? What steps can be taken to investigate the computational efficiency of the method and explore potential optimizations or approximations to improve its practical applicability?

How robust is the TV score method to adversarial attacks, and what are the critical considerations for the security of generative language models? What steps can be taken to evaluate the method's performance under different types of adversarial perturbations and develop strategies to improve its robustness?

---

> ### Author Rebuttal · Authors · 2024-08-05
>
> **Thanks for your constructive comments! We will respond to your concerns one by one.**
>
> ---
>
> > W1 & Q1: The mechanism of TV score ... more analysis and visualization of trajectory ... intuition behind the choice of trajectory volatility ...
>
> ### 1. Mechanism of TV score and its relationship to "pattern collapse"
>
> The relationship of TV score and "pattern collapse" are not directly related, but indirectly transitioned through the theoretical intuition of Section 2.1. Refer to **"General Rebuttal: Motivation line from "pattern collapse" to "early stabilization" and TV score"** for details.
>
> ### 2. Intuition behind the choice of trajectory volatility
>
> The Intuition is our Hypothesis 1 (line 104): The "pattern collapse" in mathematical reasoning scenarios leads to more significant differences among different samples' trajectories compared to traditional generative tasks (we have modeled and proved this intuition from a theoretical perspective in Section 2.1)
>
> ### 3. Detailed analysis and visualization of trajectory dynamics
>
> Refer to **"General Rebuttal: Universality of "Early Stabilization" phenomenon in mathematical reasoning (Expanded visualization of Figure 3 in the paper)"** for detailed responses.
>
> ---
>
> > W2 & Q2: The computational complexity is not discussed ...
>
> Thanks for your consideration. After obtaining all outputs $\boldsymbol{y_l}$ of each layer, there are two main steps to obtain the final scores:
>
> (1) Get ID Data Information: Fitting Gaussian distribution $\mathcal{G}_l$ = $\mathcal{N}(\boldsymbol{\mu}_l, {\boldsymbol \Sigma}_l)$ for $L$ layers
>
> ${\boldsymbol \Sigma}_l$ is a diagonal matrix because $d$ embedding dimensions are independent, so we only need to compute the mean and variance of each dimension of all $n$ ID sample embeddings.
>
> * Compute mean ${\boldsymbol{\mu}}_l$: $d(n-1)$ addition operations and $d$ multiplication operations are required;
> * Compute variance ${\boldsymbol \Sigma}_l$: $dn+d(n-1)=d(2n-1)$ addition operations and $dn+d=d(n+1)$ multiplication operations are required.
>
> We fit $L$ Gaussian distributions, so the number of addition and multiplication operations are both $\mathcal{O}(Ldn)$.
>
> We report the computation time of fitting Gaussian distribution in the ID dataset MultiArith ($n$=600):
> | time (s) |
> | --- |
> | 1.132 |
>
> *Note: this part is a one-time event and only needs to be performed once.*
>
> (2) Get OOD Data Information: Compute TV Score
>
> For $k=0$, we need to compute the Mahalanobis Distance between OOD sample and ID distribution (Eq 6). Since ${\boldsymbol \Sigma}_l$ is a diagonal matrix, it involves only simple vector multiplication and does not require matrix multiplication. Specifically, it requires $d + (d-1) = 2d-1$ addition operations and $2d$ multiplication operations. We have $L$ layers, so the numbers of addition and multiplication operations are both $\mathcal{O}(Ld)$.
>
> For $k>0$, for each increase in $k$ by 1, $L$ Gaussian distribution differences, $L$ embedding differences, and $L$ MD computations need to be performed. The increasing numbers of addition and multiplication are both $\mathcal{O}(Ld)$.
>
> We sample 1000 cases to compute TV score. Time is as below:
>
> |k|time mean (s)|time std (s)|
> |-|:-:|:-:|
> |0|0.0016|0.0001|
> |1|0.0033|0.0001|
> |2|0.0049|0.0002|
> |3|0.0066|0.0002|
> |4|0.0082|0.0002|
> |5|0.0098|0.0003|
>
> **After complexity analysis and experimental results, it is clear that our method is efficient and can be flexibly deployed in realistic scenarios.** *This is one of our strengths, especially compared to some probability-based metrics such as perplexity, which require time-consuming softmax exponential computation.*
>
> ---
>
> > W3 & Q3: The paper does not address the robustness ...
>
> Thanks for your consideration. However, prior works on OOD detection did not take this into account, so the robustness of OOD detection is not a well-defined question. Despite this, we try our best to design experiments to verify this point.
>
> We make the following assumption and goal: For a perturbed ID sample, the sample still belongs to the ID sample in realistic scenarios, but models may misidentify it as an OOD sample. We need to avoid this misidentification.
>
> We refer to [1] for perturbation method to input data in language models:
>
> * Paraphrasing: We generate one paraphrased input by querying ChatGPT using the prompt in [1];
> * Dummy Tokens: We randomly select tokens that marginally influence the original meaning and append them to the input. Such tokens could be newline characters, tab spaces, ellipses, or supplementary punctuation marks.
>
> We compare two settings:
>
> * Original: Results in our paper
> * Perturbation: We replace ID samples in the test set with the perturbed text. We report results under two perturbations.
>
> ||Llama2-7B||||GPT2-XL||||
> |-|:-:|:-:|:-:|:-:|:-:|:-:|:-:|:-:|
> ||Far-shift OOD||Near-shift OOD||Far-shift OOD||Near-shift OOD|
> ||AUROC|FPR95|AUROC|FPR95|AUROC|FPR95|AUROC|FPR95|
> |Original|98.76±0.11|5.21±0.98|92.64±0.39|28.39±1.38|93.47±0.08|24.10±0.95|94.86±0.23|13.82±0.36|
> |Perturbation w/ Paraphrasing|97.94±0.12|5.75±1.00|91.88±0.42|29.28±1.41|93.12±0.09 |24.67±0.97|94.10±0.24|15.02±0.39|
> |Perturbation w/ Dummy Tokens|98.54±0.11|5.54±0.98|92.43±0.40|29.16±1.41|93.47±0.08|24.10±0.95|95.01±0.22|12.78±0.34|
>
> **Our method can largely defend against some perturbations in realistic scenarios, showing its strong robustness.** *We conjecture that this is because our method considers a large amount of information in the middle layer, whereas probability- or embedding-based methods only consider a single layer in the output/input, which makes our method better resistant to the influence of some random factors in the output layer, such as overconfidence, and therefore more robust.*
>
> As for more explorations, we leave it for future work.
>
> [1] SPUQ: Perturbation-Based Uncertainty Quantification for Large Language Models. EACL, 2024.
>
> ---
> **We expect these responses to address your concerns, and look forward to your more positive comments.**

---

> ### Author Response · Authors · 2024-08-13
> **Thanks for reviewing & Invite feedback**
>
> Dear Reviewer SAjz,
>
> We want to express our sincere appreciation for your efforts and time spent reviewing our work and the constructive comments.
>
> During the rebuttal period, we have provided a detailed response to address all the concerns mentioned by you about **mechanism, complexity, and robustness**. With the reviewer-author discussion period ending in one day, we kindly invite you to give some feedback on our responses, and we really hope our responses adequately address your concerns.
>
> Best,
>
> Authors

---

> > ### Author Response · Authors · 2024-08-14
> >
> > Dear Reviewer SAjz,
> >
> > With the reviewer-author discussion period coming to an end, we really hope that we can get your feedback, and that our responses adequately address your concerns.
> >
> > Best,
> >
> > Authors

---

### Author Rebuttal · Authors · 2024-08-04

General Rebuttal: Motivation line from "pattern collapse" to "early stabilization" and TV score
---


To summarize, our motivation line is:

**(Section 1, Figure 1): We find "pattern collapse" in the output space**

**-> (Section 2.1, Theoretical Intuition and Proving): The "pattern collapse" leads to more significant trajectory differences across different samples**

**-> (Section 2.2, Empirical Experiments): The source of the trajectory differences between ID and OOD samples is the "Early Stabilization" phenomenon**

**-> (Section 3): The "Early Stabilization" makes our trajectory-based detection method effective.**

---

General Rebuttal:  Existence and Universality of "Pattern Collapse" phenomenon in mathematical reasoning
---

### 1. Why "pattern collapse"? **Tokenization is the key**

**For generative language models (GLMs), they model real numbers or mathematical expressions not in a mathematical sense, but based on discrete token sequences after tokenization. Thus, the collapse occurs at the token level, not at the full mathematical expression level.**

Due to the autoregressive generative nature of GLMs, the collapse phenomenon occurs during the prediction of each token.

### 2. More cases

For mathematical expressions that are very different in the mathematical sense, **after tokenization, they all contain only 0-9 number tokens and a limited number of special symbols**, such as decimal points, slashes, root signs, curly brackets. They make up a very small percentage of the vocab:

* 5517 -> ['▁', '5', '5', '1', '7']
* 21.59 -> ['▁', '2', '1', '.', '5', '9']
* 71/91 -> ['▁', '7', '1', '/', '9', '1']
* -\\sqrt{3255} -> ['▁', '-\\', 'sqrt', '{', '3', '2', '5', '5', '}']
* y^4-2y^3+7y^2+y-5 -> ['▁', 'y', '^', '4', '-', '2', 'y', '^', '3', '+', '7', 'y', '^', '2', '+', 'y', '-', '5']
* x^2/x^5 -> ['▁', 'x', '^', '2', '/', 'x', '^', '5']

### 3. "pattern collapse" is generally hold for all types of mathematical reasoning tasks

After explaining the tokenization behind "pattern collapse", we now demonstrate the universality of "Pattern Collapse" phenomenon in mathematical reasoning.

> Setup

To demonstrate the universality of “pattern collapse” across various tasks of mathematical reasoning, we conduct the following statistical experiment:
We **categorize the mathematical tasks into various types across different domains and difficulties, then count the token type number and token duplication rate corresponding to each category, and the vocab coverage**. We also test translation and summarization tasks by taking samples with the same token size as the mathematical reasoning dataset for a clear comparison.

We use Llama2 tokenizator (Vocab size = 32000). The computation metric is:

* Token Duplication Rate = 1 - token type number / token number

* Vocab Coverage = token type number / Vocab size

> Statistics Data

***{The statistics data comparisons are shown in PDF Table 2.}***

> Analysis

From the results, we can conclude that:

* Existence: The **average token duplication rate is up to 99% on all math tasks**, and even **a staggering 99.9% on some simple arithmetic tasks**; In contrast, the **token duplication rate on the text generation task is only about 60%, with about 2000 different types of token**, and still increasing as the total number of tokens increases. These data and comparisons demonstrate that pattern collapse occurs in mathematical reasoning and not in text generation.

* Universality: Token repetition rate **exceeded 97% on all seven math tasks** of different difficulties and types.

> Conclusion

These evidences demonstrate that the **"pattern collapse" occurs on generally all types of mathematical reasoning tasks**.

---

General Rebuttal: Universality of "Early Stabilization" phenomenon in mathematical reasoning (Expanded visualization of Figure 3 in the paper)
---
> Setup

**We present detailed visualization of the "early stabilization" phenomenon (detailed version of Figure 3 in the paper), which contains comparisons of trajectory volatility (embedding differences between neighboring layers) curves and sample standard deviation (color shading) between ID and OOD samples.**

The language model is Llama2-7B (32 layers, so 31 neighboring layer-pairs).  The ID data is the MultiArith dataset (Arithmetic domain, primary difficulty), and the OOD data consists of 10 datasets from different tasks and difficulties (identical to the setup in Section 4.1):

* Near-shift OOD:  GSM8K, SVAMP, AddSub, SingleEq, SingleOp (Arithmetic domain;  middle-school difficulty)
* Far-shift OOD: MATH-Algebra/Geometry/Counting_and_Probability/Number_Theory/Precalculus (algebra, geometry, counting-and-prob, number-theory, precalculus domain); university difficulty)

> Visualization and Corresponding Statistics Data

***{The visualization figures of all datasets are shown in PDF Figure 1.}***

***{The average volatility statistics for layers 1-31 (full layers), 20-31, and 26-31 on each dataset are shown in PDF Table 1.}***


> Analysis

From the detailed visualizations and statistical data, we can conclude that:

* Dataset Level:
    * ID samples all show the "Early Stabilization" phenomenon compared to OOD math problems of different types and difficulties, because the average volatility in 26-31 layers of the ID dataset is significantly lower than that of OOD datasets;
    * The mid-to-late volatilities on the far-shift OOD dataset are more dramatic due to the greater deviation from the distribution of the ID dataset.
* Sample Level: Trajectory volatility can vary somewhat across samples, but in general, ID samples have less trajectory volatility than OOD samples, and in the mid-to-late layers, ID samples generally complete the main inference, while OOD samples still have large learning volatility overall.

> Conclusion

**These detailed visualizations and analyses can demonstrate the generalizability of the "early stabilization" phenomenon in mathematical reasoning scenarios.**

---

### Author Response · Authors · 2024-08-10
**Thanks for reviewing & Invite feedback**

Dear Reviewers,

We want to express our sincere appreciation for your efforts and time spent reviewing our work and the insightful and constructive comments.

During the rebuttal period, we have provided a detailed response to address all the concerns mentioned by all reviewers.
With the reviewer-author discussion period half over, we kindly invite you to give some feedback on our responses, and we really hope our responses adequately address your concerns.

Best,

Authors

---

### Decision · Program_Chairs · 2024-09-25

**Decision:**

Accept (poster)

**Comment:**

This paper proposes a OOD detection method for large language models in mathematical reasoning. The problem of OOD detection in the context of mathematical reasoning tasks has not been investigated although robust OOD detection is important for real-world deployment of LLMs. The originality of the problem setting is high. The proposed method based on trajectory volatility is reasonable. The extensive experimental results demonstrate the effectiveness of the proposed method well. There are several points that should be improved in this paper, which include the mathematical writing, and clarity of the motivation. The paper should be revised incorporating the results presented in the rebuttal and the discussion.